# *wtf* genes are prolific dual poison-antidote meiotic drivers

**Nicole L Nuckolls[1†], María Angélica Bravo Núñez[1†], Michael T Eickbush[1], Janet M Young[2], Jeffrey J Lange[1], Jonathan S Yu[2‡], Gerald R Smith[2], Sue L Jaspersen[1,3], Harmit S Malik[2,4]*, Sarah E Zanders[1,3]***

[1]Stowers Institute for Medical Research, Kansas City, United States; [2]Division of Basic Sciences, Fred Hutchinson Cancer Research Center, Seattle, United States; [3]Department of Molecular and Integrative Physiology, University of Kansas Medical Center, Kansas City, United States; [4]Howard Hughes Medical Institute, Fred Hutchinson Cancer Research Center, Seattle, United States

**Abstract** Meiotic drivers are selfish genes that bias their transmission into gametes, defying Mendelian inheritance. Despite the significant impact of these genomic parasites on evolution and infertility, few meiotic drive loci have been identified or mechanistically characterized. Here, we demonstrate a complex landscape of meiotic drive genes on chromosome 3 of the fission yeasts *Schizosaccharomyces kambucha* and *S. pombe*. We identify *S. kambucha wtf4* as one of these genes that acts to kill gametes (known as spores in yeast) that do not inherit the gene from heterozygotes. *wtf4* utilizes dual, overlapping transcripts to encode both a gamete-killing poison and an antidote to the poison. To enact drive, all gametes are poisoned, whereas only those that inherit *wtf4* are rescued by the antidote. Our work suggests that the *wtf* multigene family proliferated due to meiotic drive and highlights the power of selfish genes to shape genomes, even while imposing tremendous costs to fertility.

**\*For correspondence:** hsmalik@ fredhutch.org (HSM); sez@ stowers.org (SEZ)

†These authors contributed equally to this work

**Present address:** ‡McKinsey Consulting Inc, Boston, United States

## Introduction

Infertility can be perplexingly high within eukaryotic species. For example, more than one out of every seven human couples are infertile (*Thoma et al., 2013*). This high infertility is at odds with the fundamental requirement of reproductive success for Darwinian fitness. A potential solution to this infertility paradox is the presence of selfish genes that subvert meiosis to increase their transmission into gametes (*Ségurel et al., 2011*; *Presgraves, 2010*; *Johnson, 2010*); such selfish genes might explain a subset of cases of human infertility. Gamete-killing meiotic drive alleles are one such class of selfish genes that can directly cause infertility. These genes act by killing the gametes that do not inherit them, increasing their transmission into up to 100% of the progeny of a heterozygote (*Lindholm et al., 2016*; *Sandler and Novitski, 1957*). Meiotic drivers can also indirectly result in infertility or other disease states by interfering with natural selection's ability to choose the most well-adapted alleles. Natural selection cannot harness the fitness benefits of alleles carried in gametes destroyed by drive. Furthermore, meiotic drivers can promote the spread of maladapted alleles that are genetically linked to the drive locus within a population (*Sandler and Novitski, 1957*; *Crow, 1991*). Because drive can be harmful to the overall fitness of a species, suppressors of drive often evolve in response (*Crow, 1991*).

Gamete-killing meiotic drive has been observed in eukaryotes ranging from plants to mammals (*Lindholm et al., 2016*). With the broadening implementation of high-throughput sequencing of both meiotic products and cross progeny to measure allele transmission, the presence of meiotic drive is being observed at an accelerated rate, and it is hypothesized that these selfish genes are

**eLife digest** Animals, plants and fungi produce sex cells – known as gametes – when they are preparing to reproduce. These cells are made when cells containing two copies of every gene in the organism divide to produce new cells that each only have one copy of each gene. Therefore, a particular gene copy usually has a 50% chance of being carried by each gamete. There is a group of genes that selfishly increase their chances of being transmitted to the next generation by destroying the gametes that do not carry them. These "gamete killer" genes can lead to infertility and other health problems.

Fission yeast is a fungus that is widely used in research. Previous studies revealed that the yeast are likely to have several gamete killers, but the identities of these genes or how they work were not clear. Nuckolls, Bravo Núñez et al. sought to identify at least one gamete killer gene and understand how it works.

The experiments found that a gene called *wtf4* acts as a gamete killer in fission yeast. This gene encodes two different proteins, one that acts as a poison and one that acts as an antidote. The antidote remains inside the gametes that contain the *wtf4* gene, while the poison is released in the surrounding environment. The poison is capable of killing all of the gametes, but the antidote protects the gametes that contain the *wtf4* gene. Further experiments show that *wtf4* is just one member of a large family of genes that are also likely to play roles in selectively killing gametes.

A separate study by Hu et al. found that two other members of the *wtf* family also act as gamete killers in fission yeast. Together, these findings expand our understanding of the nature of gamete killers and how they can contribute to infertility. This may guide the search for gamete killers in humans and other organisms. In the future, gamete killers could potentially be used to eradicate populations of pests that damage crops or spread diseases in humans.

common (*Lindholm et al., 2016*; *Didion et al., 2015*; *Ottolini et al., 2015*; *Grognet et al., 2014*; *Burt and Trivers, 2006*). However, only a handful of genes involved in meiotic drive have been identified. Lack of homology among these genes makes it nearly impossible to identify novel drive loci from genome sequences alone. Instead, rigorous genetic analyses are required to detect and map meiotic drive loci. These efforts are frequently impeded by the complexity of many drive systems; they often have multiple components and are found within chromosome rearrangements that are recalcitrant to genetic mapping (*Larracuente and Presgraves, 2012*; *Bauer et al., 2012*). Even in the case of well-studied meiotic drive systems where one or more components have been identified, a complete understanding of the mechanistic basis of drive and its suppression has been elusive (*Grognet et al., 2014*; *Larracuente and Presgraves, 2012*; *Bauer et al., 2007*, *2005*; *Hammond et al., 2012*).

The prospect of characterizing meiotic drivers in a genetically tractable system spurred our study of a pair of fission yeasts, *Schizosaccharomyces pombe* strain 972 (*Sp*) and *Schizosaccharomyces kambucha* (*Sk*). Despite being 99.5% identical at the nucleotide level, *Sp/Sk* hybrids are nearly sterile (*Rhind et al., 2011*; *Zanders et al., 2014*); a reproductive barrier between these yeasts must have arisen very recently. This rapid evolution of infertility is common amongst fission yeasts that are generally categorized as isolates of the *Schizosaccharomyces pombe* species (*Avelar et al., 2013*). In the case of *Sp/Sk* hybrids (and likely other pairings), the infertility is caused by both chromosomal rearrangements and multiple meiotic drivers (*Zanders et al., 2014*; *Avelar et al., 2013*). Indeed, we previously found that genes on each of the three *Sk* chromosomes are capable of enacting gamete (spore)-killing meiotic drive against their *Sp* homologs (*Figure 1A*) (*Zanders et al., 2014*). However, the genes responsible for the drive phenotypes were unknown.

Here, we use genetic mapping to identify *Sk wtf4* as an autonomous gamete-killing meiotic drive gene. We show that *Sk wtf4* generates two transcripts from alternative start sites: a long transcript encoding an antidote and a short transcript encoding a gamete-killing poison. Whereas the poison protein is found in all the gametes, the antidote protein is enriched only in the gametes encoding *Sk wtf4*, thereby ensuring that gametes that do not inherit the selfish allele are destroyed. This gene is a member of the large, rapidly evolving *wtf* gene family that has 25 members in *Sp*. We show that

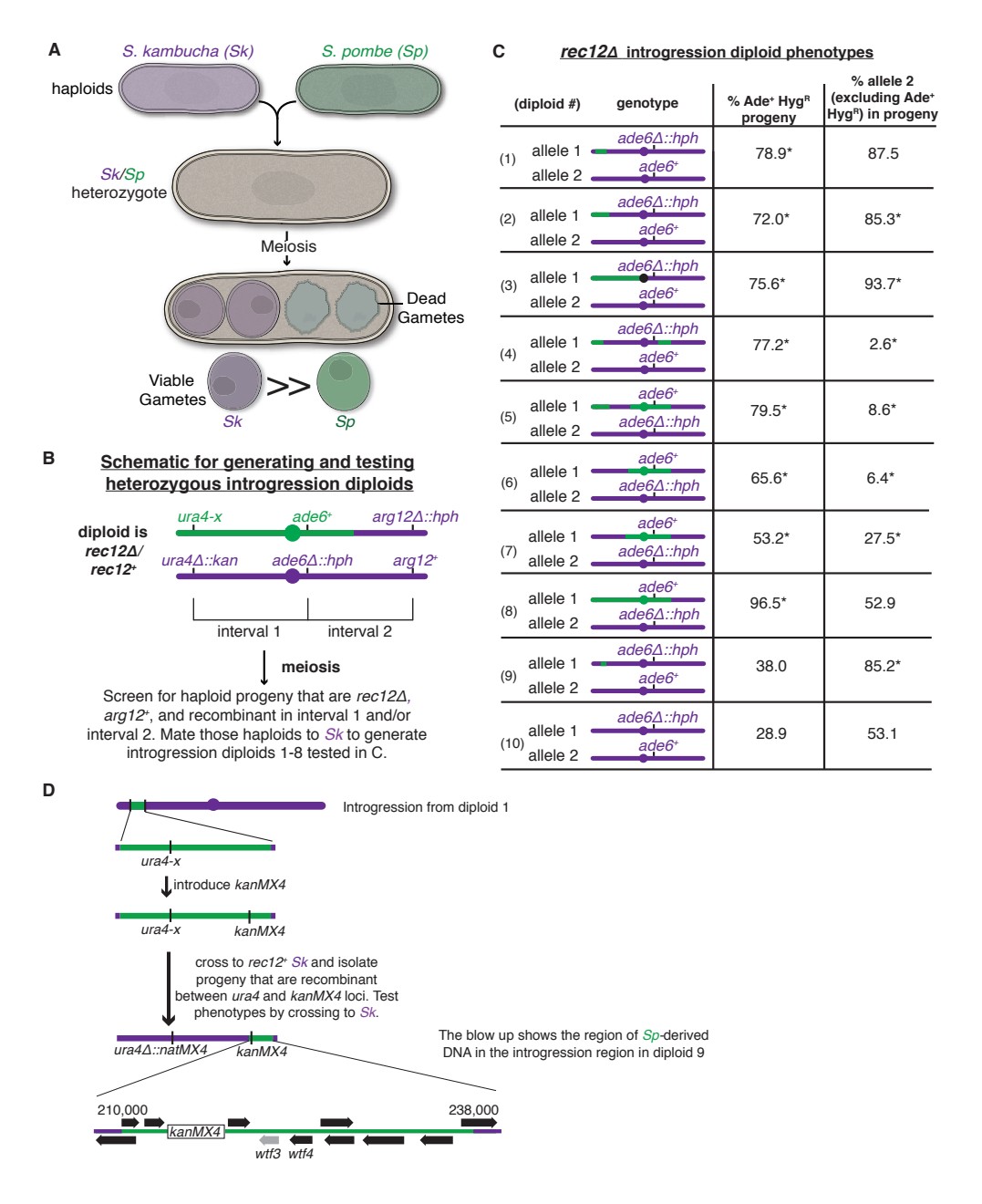

**Figure 1.** A complex meiotic drive landscape on *Sk* and *Sp* chromosome 3 is revealed by recombination mapping. (**A**) A cross between *Sk* and *Sp* generates a heterozygote that has low fertility and preferentially transmits *Sk* alleles on all three chromosomes into viable gametes (*Zanders et al., 2014*). (**B**) Generation of chromosome 3 introgression diploids 1–8. *Sk*-derived DNA is shown in purple while *Sp*-derived DNA is shown in green. The origin of the *Sp/Sk* mosaic chromosome is depicted in *Figure 1—figure supplement 1*. (**C**) Phenotypes of *rec12Δ/rec12Δ* introgression/*Sk* diploids. See *Figure 1—source data 1* for breakpoints between *Sk*-derived DNA (purple) and *Sp*-derived DNA (green). Chromosome transmission was followed using the heterozygous markers at the *ade6* locus: *hph* is short for the *hphMX4* marker gene which confers resistance to hygromycin (Hyg[R]). The percentage of gametes that inherit both markers (heterozygous disomes, likely aneuploids and diploids) and (after excluding the heterozygous disomes) the percent of gametes that inherit the marker from the pure *Sk* chromosome are shown. Over 100 viable gametes were tested for each diploid; raw data can be found in *Figure 1—source data 2*. * indicates p-value<0.01 (G-test) compared to *rec12Δ/rec12Δ Sk* control (from *Zanders et al. (2014)*). (**D**) Fine-scale mapping of the drive locus starting with the introgression from diploid 1. Strains that were recombinant between the *ura4* locus and an introduced *kanMX4* marker gene were selected and their phenotypes were tested in crosses to *Sk*. The recombinant strain with the smallest amount of *Sp* DNA that retained the phenotype (sensitivity to drive by an *Sk* chromosome) is shown in detail. This introgression strain was mated to *Sk* to generate diploid 9. These analyses identified a ~30 kb candidate region (see blow up) containing a drive locus. In *Sp*, this region contains *wtf4* and the *wtf3* pseudogene. The syntenic region in *Sk* contains only one *wtf* gene, *wtf4*.

*Figure 1 continued on next page*

*Figure 1 continued*

The following source data and figure supplement are available for figure 1:

**Source data 1.** Breakpoints between *Sp* and *Sk*-derived DNA sequences.
**Source data 2.** Raw data underlying *Figure 1C*.
**Figure supplement 1.** Generation of mosaic chromosome 3 used in *Figure 1B*.

*wtf4* is not the only driver amongst *wtf*s and propose a model in which meiotic drive is the ancestral function of the gene family. Our study thus identifies a novel mechanism by which meiotic drivers can act and highlights the significant role these selfish elements have played in shaping the evolution of a model eukaryote.

## Results

### Genetic mapping reveals a complex landscape of drive loci and modifiers

To study meiotic drive in fission yeast, we mate haploids to generate diploids, induce the diploids to undergo meiosis and monitor allele transmission into the gametes using genetic markers. In *Sk/Sp* hybrid diploids, drive of loci on all three *Sk* chromosomes is due to the preferential death of gametes inheriting the corresponding *Sp* alleles (*Zanders et al., 2014*) (*Figure 1A*). In this work, we focused on chromosome 3 because it is the smallest chromosome and the drive phenotype is strong: greater than 80% of viable haploid gametes inherit an *Sk* marker allele from *Sk/Sp* hybrids (*Zanders et al., 2014*).

To genetically map a drive locus on chromosome 3, we first wanted to generate a strain with *Sk* chromosomes 1 and 2, but *Sp* chromosome 3. Because *Sp* and *Sk* have different karyotypes on chromosomes 2 and 3 due to a translocation (*Zanders et al., 2014*), we could not generate such a strain as it would lack essential genes. Instead, we generated a haploid strain with an *Sk* karyotype containing *Sk* chromosomes 1 and 2 and most, but not all, of chromosome 3 derived from *Sp* (*Figure 1—figure supplement 1* and Materials and methods). We then backcrossed this haploid strain to *Sk* to generate a series of haploid strains that have mosaic (*Sp* and *Sk*-derived DNA sequences) versions of chromosome 3 generated by recombination (*Figure 1B*). We then crossed the recombinant haploids to *Sk* to generate a series of 'introgression diploids' (*Figure 1B* and *Figure 1C*, diploids 1–8). The introgression diploids were all homozygous null mutants for *rec12*, the fission yeast ortholog of *Saccharomyces cerevisiae SPO11*, which is required for inducing DNA breaks to initiate meiotic recombination (*Phadnis et al., 2011*). As meiotic recombination is not induced in the introgression diploids, we could use any genetic marker on chromosome 3 to assay this chromosome for the presence of drive loci. We used the codominant markers *ade6+* and *ade6Δ::hphMX4* to follow transmission of each chromosome into viable gametes (*Figure 1C*).

We observed three phenotypic classes amongst our introgression diploids (diploids 1–8, *Figure 1C*). In the first class (diploids 1–3), the allele from the pure *Sk* chromosome exhibited drive over the allele from the *Sp/Sk* mosaic chromosome. In the second class (diploids 4–7), we were surprised to observe the opposite phenotype: the allele from the *Sp/Sk* mosaic chromosome exhibited drive over that from the pure *Sk* chromosome. In the third class (diploid 8), we observed unbiased allele transmission.

Our finding of three distinct phenotypic classes amongst our introgression diploids (diploids 1–8) is inconsistent with the simple model of a single drive locus on *Sk* chromosome 3. A single gene model predicts two phenotypic classes: (1) introgression diploids in which the pure *Sk* chromosome exhibits drive because the *Sk/Sp* mosaic chromosome lacks the *Sk* drive allele and (2) introgression diploids in which the chromosomes show Mendelian transmission because the *Sk/Sp* mosaic contains the *Sk* drive allele.

Instead, our data are more consistent with the presence of a meiotic drive allele (or alleles) found on both *Sk* and *Sp* chromosome haplotypes and the existence of at least one genetically separable drive suppressor. The drive of the *Sk/Sp* mosaic chromosome over the pure *Sk* chromosome in class 2 (diploids 4–7) is consistent with the presence of an *Sp* drive allele in these strains. The full effects of this *Sp* drive locus could have been missed previously in *Sk/Sp* hybrid crosses due to the actions of an *Sp* drive suppressor not found in the class 2 introgressions (*Zanders et al., 2014*).

Similar to what we previously observed in crosses between pure *Sk/Sp* hybrids (both $rec12^+$ and $rec12\Delta$), we found that viable gametes produced by diploids of all three classes frequently inherited both alleles at the *ade6* locus (*Figure 1C*) (*Zanders et al., 2014*). This indicates that they are not haploid at this locus, as is expected for gametes. These gametes likely represent a mix of heterozygous diploids and heterozygous chromosome 3 aneuploids. In diploid 8, the phenotype was extreme, with almost all the viable gametes inheriting both *ade6* alleles (*Figure 1C*). Although the frequency of meiotic chromosome missegregation is elevated in $rec12\Delta$ mutants (*Phadnis et al., 2011*), we see significantly higher levels of viable gametes that inherit both alleles in diploids 1–8 than we did in a homozygous *Sk* $rec12\Delta$ control (*Figure 1C*, diploid 10).

The high level of chromosome 3 aneuploidy and/or diploidy we observe in the viable progeny of *Sk/Sp* hybrid crosses and our introgression diploids (1-8) is also consistent with the existence of both *Sk* and *Sp* active meiotic drive loci. We previously showed in *Sk/Sp* hybrids that this phenotype was not due to elevated chromosome missegregation in meiosis, but rather preferential death of haploid gametes (*Zanders et al., 2014*). As we proposed previously, this phenotype could result from distinct competing *Sk* and *Sp* driver loci on chromosome 3 (*Zanders et al., 2014*; *Bomblies, 2014*). In the absence of recombination, a given haploid gamete can inherit only the *Sk* or *Sp* drive locus and is thus sensitive to being killed by the one it does not inherit. Heterozygous diploids and heterozygous aneuploids, however, would inherit both loci and be resistant to both killers.

To map driver location(s) from the phenotypic data described above, we sequenced the haploid strains that contributed the *Sk/Sp* mosaic chromosomes to the introgression diploids (diploids 1–8) and combined genotype information with the phenotypic data described above. We determined which regions of chromosome 3 were derived from *Sk* and which were from *Sp* in each strain (*Figure 1C* and *Figure 1—source data 1*). It was clear from our data that one or two loci are not sufficient to explain the phenotypes of all these strains.

We chose to focus on the *Sk/Sp* mosaic chromosome found in diploid 1. This strain has the smallest amount of *Sp* DNA (~180 kb), and drive of *Sk* in the introgression/*Sk* diploid suggested the strain lacks a drive allele found in *Sk* (*Figure 1C*). We crossed a haploid isolate containing this chromosome to a $rec12^+$ *Sk* strain to generate recombinant progeny containing smaller segments of *Sp*-derived DNA (*Figure 1D* and Materials and methods). We SNP-genotyped those recombinants and tested their phenotypes by mating them to *Sk* to generate additional introgression diploids (see Materials and methods). We selected diploid 9 for further analysis, as it contains the *Sk/Sp* mosaic chromosome with the smallest region of *Sp*-derived DNA (~30 kb) that a pure *Sk* chromosome can drive against (*Figure 1C* and *Figure 1D*). After excluding aneuploid/diploid progeny (those that inherit both *ade6* markers), the allele from the pure *Sk* chromosome shows essentially the same transmission bias in diploids 1 and 9. These results suggest the *Sk* drive allele active in diploid 1 is found in this ~30 kb region. Curiously, this locus is in a region that is transmitted in a Mendelian manner (to ~50% of progeny) in pure *Sk/Sp* hybrids (*Zanders et al., 2014*), suggesting that other loci can mask the effects of the driver within this ~30 kb region. In addition, it is unclear why the fraction of viable progeny that inherit both *ade6* alleles drops between diploids 1 and 9. These puzzles likely reflect the complexity of the multiple drivers and suppressor loci acting in these yeasts (*Zanders et al., 2014*).

We next wanted to verify the candidate drive locus using a recombination-competent ($rec12^+$) diploid. We generated introgression diploid 11 which contains the same *Sk/Sp* mosaic chromosome as diploid 1, but is $rec12^+$. To follow the transmission of the candidate locus, we needed a closely linked marker gene, so we engineered heterozygous markers at the linked *ura4* locus (*Supplementary file 1*). We found that the *ura4* allele from the pure *Sk* chromosome is transmitted to 87% of the viable gametes produced by diploid 11, which is not significantly different from the 88% transmission of the *Sk* allele in diploid 1 (*Figure 1C* and *Figure 2A*). This result shows that *ura4* is closely linked to an *Sk* drive locus and is consistent with that locus being within the ~30 kb candidate region.

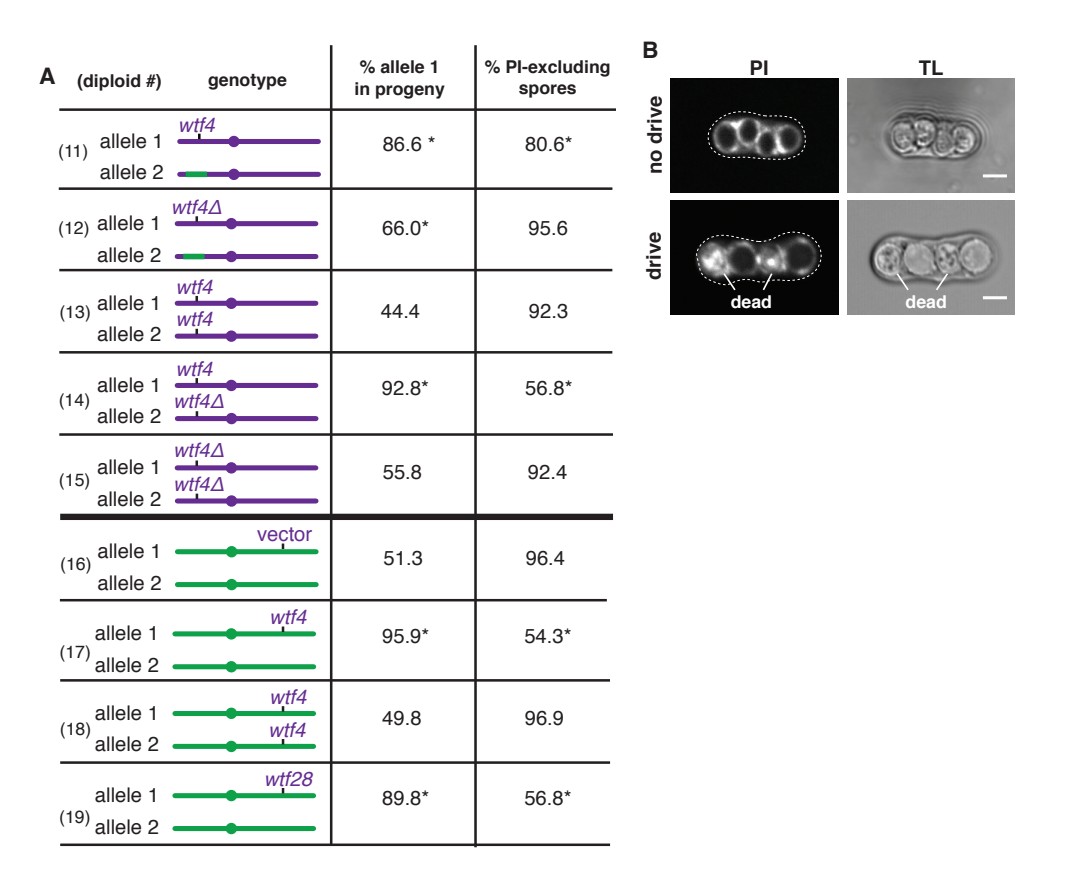

**Figure 2.** *Sk wtf4* is a self-sufficient meiotic driver that kills gametes that do not inherit the gene. (A) Allele transmission and propidium iodide (PI) staining phenotypes of diploids 11–19. *Sk*-derived DNA is purple, *Sp*-derived DNA is green. The cartoons depict chromosome 3. Chromosomes 1 and 2 are derived from *Sk* in diploids 11–15, but are from *Sp* in diploids 16–19. For diploids 11–15, allele transmission was monitored by following heterozygous markers at the *ura4* locus, which is tightly linked to *wtf4* (estimated 7–17 cM based on physical distance [*Young et al., 2002*]). PI dye is excluded from living spores, but not dead spores that have lost membrane integrity, such as those destroyed by drive. The percent of spores that exclude PI is shown as a proxy of fertility (*Figure 2—source data 1*). The PI phenotypes and *ura4* locus allele transmission for diploids 11, 12, 14 and 15 were compared to those of the wild-type *Sk* control (diploid 13). * indicates p-value<0.01 (G-test). For diploids 16–19, allele transmission was followed using markers at the *ade6* locus, which is where the empty vector or *wtf* gene constructs are integrated. The integrations introduced a dominant drug resistance gene and mutated *ade6*[+]. Because these diploids all had codominant alleles at *ade6*, we could detect progeny that inherited both *ade6* alleles (less than 10% of the total population). These progeny are excluded from the data presented above, but all the raw data are presented in *Supplementary file 1*. The PI phenotypes and allele transmission for diploids 17–19 were compared to the empty vector control (diploid 16) and * indicates p-value<0.01 (G-test). See *Supplementary file 1* for the markers used for each diploid and the raw data for allele transmission and *Supplementary file 2* for the PI staining raw data. Over 200 viable gametes were scored for allele transmission and over 200 spores (>50 4-spore asci) were assayed for PI staining. (B) Images of PI staining and transmitted light (TL) in an ascus with no drive containing all alive spores (top) and in an ascus with drive where two of the four spores are dead (bottom). Scale bar represents three microns.

The following source data is available for figure 2:

**Source data 1.** PI staining correlates with viable spore yield as a measure of fertility in wild-type and *wtf* heterozygous crosses.

To test whether the transmission bias we observed in diploid 11 might be caused by increased cell death amongst gametes inheriting the *Sp* locus, we used propidium iodide (PI) to stain the meiotic sacs (asci) that hold the spores. PI efficiently stains dead cells that have lost their membrane integrity but fails to stain viable cells (*Figure 2B* and *Figure 2—source data 1*) (*Moore et al., 1998*). We found that only 81% of spores generated by diploid 11 excluded PI, while wild-type strains (e.g. diploid 13) have rates >90% (*Figure 2A*). Together, our findings support the hypothesis that the *Sk* ~30 kb region encodes a gamete-killing meiotic driver.

## *Sk wtf4* is a meiotic drive locus

Near the center of the *Sk* 30 kb candidate region is *wtf4* (*Figure 1D*), a member of the mostly uncharacterized *wtf* gene family. This family contains 25 members in *Sp*, and its (cheeky) name is derived from the genes' genomic association <u>w</u>ith <u>Tf</u> transposons (*Bowen et al., 2003*). *wtf* genes are not found outside *Schizosaccharomyces* species (*Bowen et al., 2003*). *Sk wtf4* is a 1427 bp gene (from the start to stop codon, including introns) with six exons and encodes a protein with six predicted transmembrane domains. *Sk wtf4* shares only 89% DNA sequence identity (82% amino acid identity) with the gene in the orthologous locus in *Sp* (*Sp wtf4*); this divergence is much higher than expected given the 99.5% average DNA sequence identity between the two genomes (*Rhind et al., 2011*; *Zanders et al., 2014*). We reasoned that *wtf* genes, in general, were good candidates for meiotic drive loci because of their rapid evolution and their transcription during meiosis (*Bowen et al., 2003*; *Mata et al., 2002*; *Daugherty and Malik, 2012*; *McLaughlin and Malik, 2017*).

To test if *Sk wtf4* is a meiotic drive gene, we deleted *Sk wtf4* (*Sk wtf4Δ*) in a pure *Sk* background and mated that haploid to one containing the same *Sk/Sp* mosaic found in diploid 11 (*Figure 2A*) to produce diploid 12. We observed a significant increase in the number of spores that could exclude PI in diploid 12 (*Sk wtf4Δ*), compared to diploid 11 (*Sk wtf4⁺*) from 81% to 96%, suggesting *Sk wtf4⁺* promotes spore death in progeny of heterozygous diploids. In addition, *Sk wtf4Δ* showed more equitable allele transmission. While *Sk wtf4⁺* is transmitted to 87% of the viable gametes produced by diploid 11, the transmission rate of *Sk wtf4Δ* is reduced to 66% in diploid 12 (*Figure 2A*). Although some residual transmission bias remains in this background, our results clearly implicate *Sk wtf4* as a large contributor to gamete-killing meiotic drive.

## *Sk wtf4* drive is consistent with a poison-antidote mechanism

There are two known means by which gamete-killers act to eliminate competing alleles (*Lindholm et al., 2016*; *McLaughlin and Malik, 2017*). Under one model, meiotic drivers kill gametes containing a particular target locus. For example, the *Segregation Distorter* (*SD*) system in *Drosophila melanogaster* kills sperm bearing an expansion of the *Responder* satellite DNA (*Larracuente and Presgraves, 2012*; *Wu et al., 1988*). The second model is a poison-antidote model in which a gamete-killing entity (the poison) is encoded at a position that is closely linked to that encoding a second substance (the antidote) which specifically protects gametes that inherit the drive locus. For example, the unidentified *rfk* gene (required for killing) acts as a poison and the *rsk* gene (resistance to spore killing) acts as an antidote in the *Spore killer-2* drive locus from *Neurospora intermedia* (*Hammond et al., 2012*; *Harvey et al., 2014*).

We first tested if *Sk wtf4* acts analogously to *SD* to kill gametes that inherit a particular *Sp* chromosomal locus. To test this idea, we analyzed the effect of *Sk wtf4Δ/Sk wtf4⁺* heterozygosity in a pure *Sk* strain background (diploid 14, *Figure 2A*). As this *Sk wtf4Δ/Sk wtf4⁺* heterozygote contains no *Sp* DNA, there should be no drive if *wtf4* can only target and drive against *Sp* sequence. We did, however, observe strong drive (93% transmission) of *Sk wtf4⁺* relative to *Sk wtf4Δ* in diploid 14 and a concomitant decrease in the percent of spores that could exclude PI (59% versus 92% in wild-type; *Figure 2A*, diploids 14 and 13). These results demonstrate that the drive of *Sk wtf4* does not require an *Sp* target sequence.

Our results are, however, consistent with a poison-antidote model of meiotic drive. The phenotype of the *Sk wtf4Δ/Sk wtf4⁺* heterozygote (*Figure 2A*, diploid 14) suggests that *Sk wtf4* acts as the antidote because gametes lacking the gene die. If this were true and a separate gene acted as the poison, we predicted that *Sk wtf4Δ* homozygotes (diploid 15) should have very low fertility because they would generate a poison, but no antidote. Contrary to this expectation, we found that an *Sk wtf4Δ* homozygote is healthy, with the same ability to exclude PI from the spores as wild-type *Sk*

(92% of spores; *Figure 2A*, diploids 13 and 15). This finding rules out the possibility that *Sk wtf4* encodes a gene important for meiosis or spore development. Instead, our results suggest that *Sk wtf4* acts as *both* poison and antidote, similar to the *Spok* genes of *Podospora anserina* (*Grognet et al., 2014*).

It remains unclear, however, why the phenotype of *Sk wtf4* is slightly weaker in the hybrid background (assayed in diploids 11 and 12) compared to the phenotypes in pure *Sk* (diploids 13–15) or pure *Sp* (diploids 16–18). We speculate it could be due to the composition (chromatin state or a sequence variant) of the mosaic chromosome (allele 2 in diploids 11 and 12).

To further test the idea that *Sk wtf4* encodes an autonomous poison-antidote drive locus, we moved the gene to a naive genome and tested if it could induce drive. We integrated *Sk wtf4* into the *Sp* genome at the *ade6* locus, which is unlinked to the endogenous *wtf4* locus. An *Sp* diploid that is hemizygous for *Sk wtf4* (*Sk wtf4+/ade6+*) produced fewer viable spores (54% PI-excluding spores, versus 96% in the vector-only control) and showed a marked transmission bias (96%) favoring *Sk wtf4+* (*Figure 2A*, diploids 16 and 17). In contrast, *Sp* diploids homozygous for *Sk wtf4+* produced viable spores that excluded PI at the same frequency as spores from wild-type diploids and showed unbiased allele transmission (*Figure 2A*, diploids 18 and 16). These results are consistent with *Sk wtf4* acting as a complete one-gene poison-antidote drive system that causes the death of gametes that fail to inherit the locus from heterozygotes.

## *Sk wtf4* generates a poison and an antidote from alternate transcripts

We hypothesized that *Sk wtf4* encodes two products to achieve drive (*Figure 3A*). The first of these is a gamete-killing poison, which acts indiscriminately on all spores. The second product is an antidote that specifically rescues only the gametes encoding *Sk wtf4* from the poison. To investigate how *Sk wtf4* could make two products, we analyzed long-read sequence data from *Sp* meiotic mRNAs (*Kuang et al., 2017*) (Materials and methods). This revealed that *Sp wtf4* is transcribed during meiosis and generates two major overlapping transcripts with different start sites (*Figure 3—figure supplement 1*). Since the region starting 500 bp upstream of the annotated *Sp wtf4* start codon until the putative second start codon is fairly well conserved (98% identical) between *Sp* and *Sk wtf4*, we hypothesized that *Sk wtf4* is likely to produce similar alternate isoforms to *Sp wtf4*. These alternative transcripts of *Sk wtf4* could encode the two meiotic drive components – a poison and an antidote (*Figure 3B*).

To test the feasibility of our model, we investigated the localization of *Sk* Wtf4-GFP in *Sp* diploids induced to undergo meiosis (*Sheff and Thorn, 2004*). We C-terminally tagged the gene to visualize proteins generated by both the putative *Sk wtf4* isoforms; this tag does not interfere with *Sk wtf4's* ability to function as a drive allele (see data for 'GFP diploid' in *Supplementary files 1* and *2*). Visualizing *Sk wtf4-GFP/ade6+* heterozygous diploids, we observed faint cytoplasmic Wtf4-GFP signal before the first meiotic division, which intensified throughout gamete development and filled the ascus surrounding the mature gametes (*Figure 3C*). In mature asci, we observed a strong enrichment of Wtf4-GFP within only two of the four spores. We observed the same spore enrichment pattern in *Sk wtf4-GFP/Sk wtf4+* diploids in which drive does not occur (*Figure 3C*).

We hypothesized that the diffuse Wtf4-GFP localization in the ascus corresponded to the poison, whereas the enrichment within the mature spores might reflect the localization of the antidote. If this hypothesis is correct, Wtf4-GFP should be enriched in the two spores that inherit the chromosome carrying *Sk wtf4-GFP*. Consistent with this idea, we stained asci from *Sk wtf4-GFP/ade6+* diploids with PI and observed that the surviving PI-negative spores (95% of which inherit *Sk wtf4-GFP*) are indeed those with the strong Wtf4-GFP signal (*Figure 3D*; *Supplementary file 1*). The localization pattern of Wtf4-GFP is consistent with our model of *Sk wtf4* encoding two protein isoforms (*Figure 3A*).

To further test our poison-antidote model, we sought to generate alleles that could produce only the poison or only the antidote. We first mutated the start codon (ATG to TAC) that is present only in the putative short transcript. Our results (below) suggest that this mutant allele retains the antidote function but no longer functions as a poison: we therefore call this allele *Sk wtf4antidote* (*Figure 4A*). In hemizygous diploids (*Sk wtf4antidote/ade6+*), *Sk wtf4antidote* does not cause spore death (increased frequency of PI-stained spores) or the transmission bias that is observed with the wild-type *Sk wtf4* allele, suggesting that the mutant can no longer drive (compare *Figure 4B*, diploid 20 to *Figure 2A*, diploid 17). However, this allele still protects from meiotic drive since *Sk wtf4+/Sk*

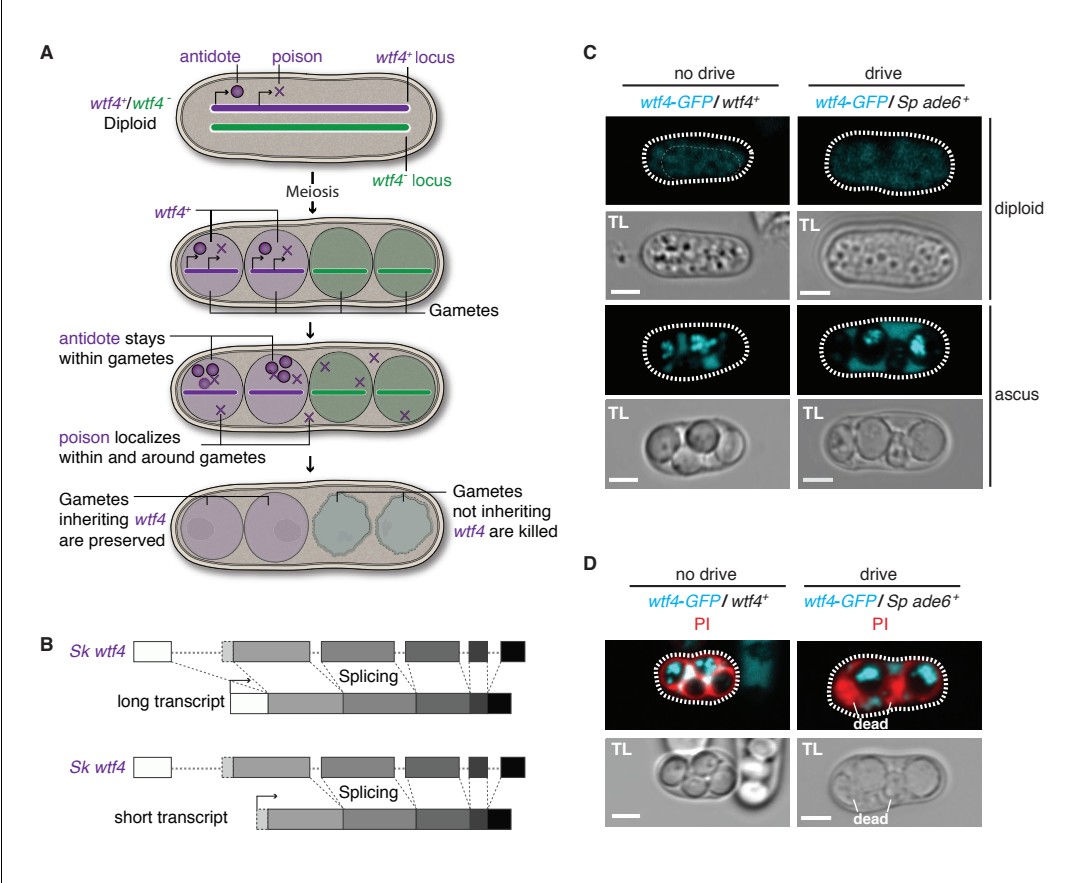

**Figure 3.** *Sk wtf4* has the capacity to make two proteins and Wtf4-GFP shows a dual localization pattern. (A) Model for meiotic drive of *Sk wtf4* via a poison-antidote mechanism. (B) *wtf4* creates a long and an alternative short transcript. See ***Figure 3—figure supplement 1*** for a depiction of the long-read RNA sequencing data on which this model is based (***Kuang et al., 2017***). (C) *Sk* Wtf4-GFP localization in diploids where drive does [right] or does not occur [left]. Cells were imaged prior to the first meiotic division [top] and as mature asci [bottom]. (D) Asci generated by diploids of the same genotypes as in (C) stained with PI to label dead cells (those lacking *wtf4*).

The following figure supplement is available for figure 3:

**Figure supplement 1.** *Sp wtf4* has alternate transcriptional start sites.

*wtf4^antidote^* heterozygotes produce PI-excluding spores at the same frequency as wild-type and show unbiased allele transmission (***Figure 4B***, diploid 21). These data assign an antidote function to the long transcript.

We next set out to generate a *Sk wtf4^poison^* allele by mutating the two putative start codons (ATG to TAG) found in exon 1 of the long transcript (***Figure 4A***). This mutant should be able to generate only the short polypeptide. If this allele retains the ability to poison spores but has lost the antidote function, we would expect all progeny to be killed in *Sk wtf4^poison^/ade6^+^* hemizygotes. Indeed, most spores generated by these diploids die (14% exclude PI stain, ***Figure 4B***, diploid 22). Interestingly, the *Sk wtf4^poison^* allele was modestly underrepresented (38% transmission) in the few surviving spores generated by diploid 22, indicating that the spores that inherit that allele are especially likely to be destroyed by their own poison (***Figure 4B***).

To confirm that the toxicity of the *Sk wtf4^poison^* allele was due to its lacking the *Sk wtf4* antidote, we generated *Sk wtf4^poison^/Sk wtf4^+^* heterozygotes. As expected, the spores that inherited the complete *Sk wtf4^+^* gene from these diploids were immune to *Sk wtf4^poison^* toxicity, while those that inherit *Sk wtf4^poison^* die. (***Figure 4B***, diploid 23). These results support our model that the short *Sk wtf4* transcript encodes a trans-acting gamete poison.

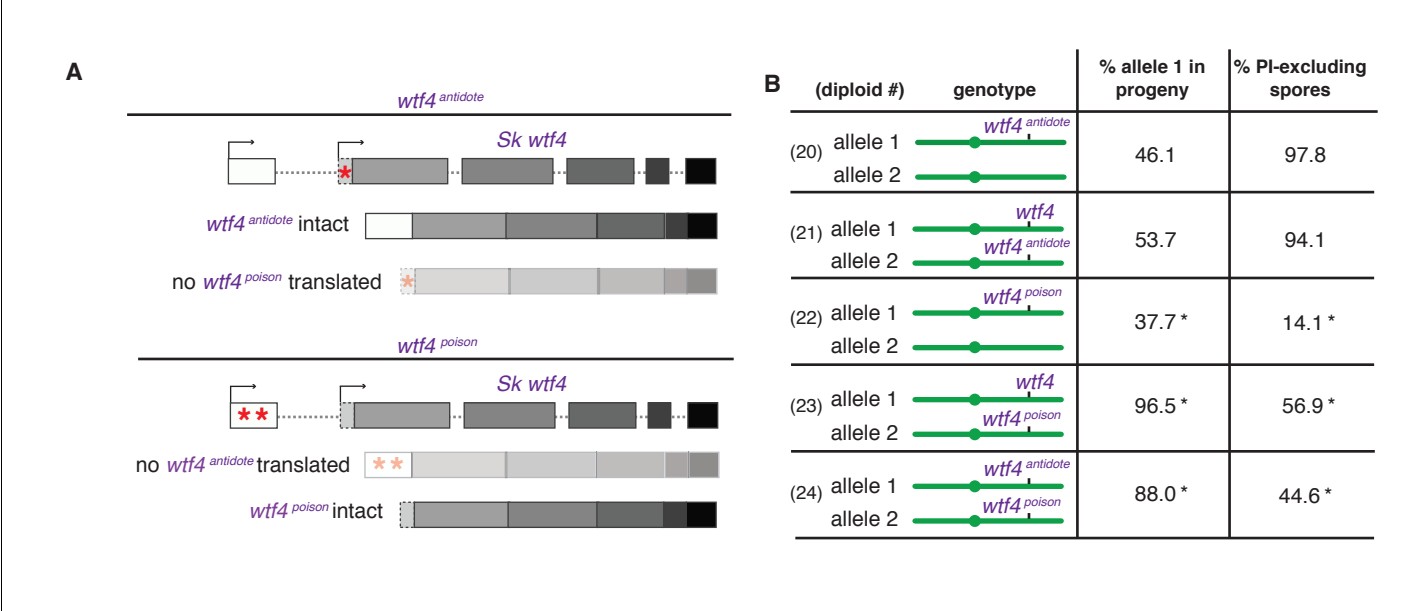

**Figure 4.** *Sk wtf4* creates two proteins using alternate transcripts: an antidote and a gamete-killing poison. (**A**) Separation of function *wtf4* alleles. The red stars indicate start codon mutations. (**B**) Allele transmission and PI staining phenotypes of *Sp* diploids with the indicated *Sk wtf4* alleles integrated at *ade6* on chromosome 3, as in diploids 16–19 in **Figure 2A**. Spores that inherited both alleles at *ade6* are eliminated from the data presented above, but the complete data are found in **Supplementary file 1**. * indicates p-value<0.01 (G-test) compared to empty vector (or wild-type control) for allele transmission and fertility as assayed by PI staining. See **Supplementary file 1** for raw data and the markers used to monitor allele transmission for each diploid and **Supplementary file 2** for the PI staining raw data. Over 200 viable gametes were scored for allele transmission for all diploids except diploid 24, from which we genotyped 50. Over 200 spores (>50 4-spore asci) were assayed for PI staining of each diploid.

As a final test of our model, we brought the separated poison and antidote mutant alleles back together in one diploid, but on opposite haplotypes. If they function as expected, we would predict that the *Sk wtf4*[poison] spores will die but the spores that inherit the *Sk wtf4*[antidote] will survive. This was indeed the case. Only 45% of the spores produced by *Sk wtf4*[antidote]/*Sk wtf4*[poison] heterozygotes can exclude PI stain and 88% of the surviving gametes inherit the *Sk wtf4*[antidote] allele (**Figure 4B**, diploid 24).

## The *Sk* Wtf4 poison is trans-acting, whereas the Wtf4 antidote is gamete-specific

We next specifically determined the localization patterns of the antidote and poison polypeptides. To visualize the antidote peptide, we generated an *Sk mCherry*[antidote]-*wtf4* allele (**Figure 5A**) and found it acts similarly to the wild-type *wtf4* allele (**Figure 5B**, diploids 25 and 26) (**Hailey et al., 2002**). We could not reliably use PI staining to assay fertility of mCherry-tagged strains because both signals are red, so we used viable spore yield assays (VSY) (**Smith, 2009**) to confirm that the fertility of the *Sk mCherry*[antidote]-*wtf4* allele was similar to untagged *wtf4* in heterozygotes. *Sk mCherry*[antidote]-*wtf4*/*ade6*[+] hemizygotes had a VSY of 0.8 ± 0.2 (standard deviation) compared to 1.0 ± 0.4 of *Sk wtf4*[+]/*ade6*[+], and *Sk mCherry*[antidote]-*wtf4*/*wtf4*[+] diploids had a VSY of 1.4 ± 0.1 compared to 1.7 ± 0.1 of wild-type.

To observe the localization of the poison peptide, we generated a *Sk wtf4*[poison]-*GFP* allele (**Figure 5A**) (**Sheff and Thorn, 2004**). While this *Sk wtf4*[poison]-*GFP* allele is not as penetrant as the untagged *Sk wtf4*[poison] allele, it does have a poison-only phenotype (**Figure 5B**, diploids 27 and 28). In *Sk mCherry*[antidote]-*wtf4*/ *Sk wtf4*[poison]-*GFP* heterozygotes, we observed *Sk* Wtf4[poison]-GFP expression before the meiotic divisions and later filling mature asci. In contrast, we observe *Sk* mCherry[antidote]-Wtf4 enriched only in two of the four mature spores (**Figure 5C**). Together, these data reconstitute the dual localization patterns we observed with *Sk* Wtf4-GFP and support our model of a poison-antidote system encoded by the same gene (**Figure 3A**).

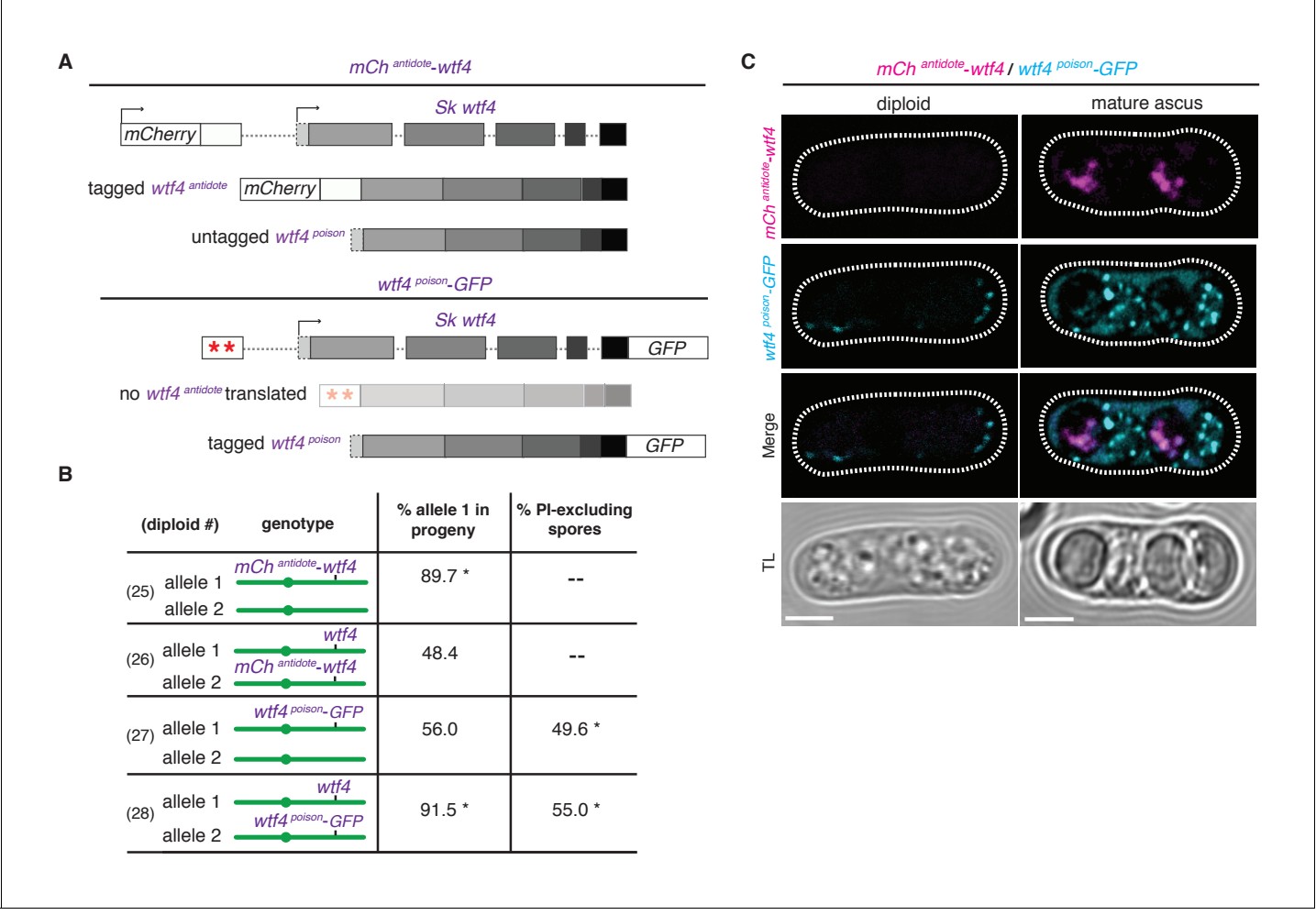

**Figure 5.** Wtf4 antidote is spore-specific and Wtf4 poison spreads throughout the ascus. (**A**) Constructs tagging either the Wtf4 antidote (top) or poison (bottom) proteins. The red stars indicate start codon mutations. (**B**) Allele transmission and PI staining phenotypes for tagged alleles, as in *Figure 4B*. See **Supplementary file 1** for raw data and the markers used to monitor allele transmission for each diploid and **Supplementary file 2** for the PI staining raw data. We could not reliably use PI to assay fertility of mCherry-tagged strains because of color similarity, but in viable spore yield assays the *mCherry*<sup>antidote</sup>-*wtf4* allele gave a similar phenotype to *wtf4*. * indicates p-value<0.01 (G-test) compared to empty vector (or wild-type control). Over 200 viable gametes were scored for allele transmission and over 200 spores (>50 4-spore asci) were assayed for PI staining. (**C**) Wtf4 poison (cyan) and antidote (magenta) protein localization prior to the first meiotic division (left) and in a mature ascus (right). Scale bar represents three microns. TL, transmitted light.

The following figure supplement is available for figure 5:

**Figure supplement 1.** Spectral unmixing verifies true signal.

## Expansion and rapid evolution of the *wtf* family is consistent with a role in meiotic drive

We hypothesized that if *Sk wtf4* is not unusual amongst the *wtf* genes in its ability to drive; meiotic drive could explain the 'driving' force behind the expansion of the *wtf* gene family (*Bowen et al., 2003*). The large number of *wtf*s could also explain the complex drive landscape revealed in our recombination mapping (*Figure 1*). To test these ideas, we analyzed additional *wtf* genes from *Sk*.

We cloned and tested six *Sk wtf* genes (*wtf2, wtf5, wtf6, wtf28,* and *wtf21* plus *wtf26* [together]) for evidence of meiotic drive. As for our tests of *Sk wtf4*, we integrated the above *Sk wtf* genes at the *ade6* locus of *Sp*, which disrupted the *ade6*<sup>+</sup> gene. We then mated those haploids to *ade6*<sup>+</sup> to generate heterozygous diploids and monitored the transmission of the *Sk wtf* gene(s) into viable

progeny using the heterozygous *ade6* markers. Five of the six genes had no observable drive phenotype. *Sk wtf2* was transmitted to 47% (n = 114) of progeny, *Sk wtf5* was transmitted to 44% (n = 454), *Sk wtf6* was transmitted to 51% (n = 471) and the combination of *Sk wtf21* and *wtf26* (cloned and integrated together) was transmitted to 46% (n = 111). However, like *Sk wtf4*, *Sk wtf28* caused strong drive (90% transmission bias and only 57% of spores excluded PI; *Figure 2A*, diploid 19).

We also compared the sequence of each of these *Sk wtf* genes to the *Sp wtf* genes at the syntenic loci. *wtf26* and *wtf28* are not found in *Sp*, so have either been lost in *Sp*, or gained in *Sk* since divergence. While *Sk wtf2* is a 1036 bp full length gene, *Sp wtf2* is likely a 388 bp pseudogene (it has a large deletion relative to other *wtf* genes and multiple in-frame stop codons). *Sk wtf21* is likely a pseudogene (multiple in frame stop codons), whereas *Sp wtf21* is intact. The two loci share 83% DNA sequence identity. The *wtf5* gene is intact in both species, and the loci share 99% DNA sequence identity and 97% amino acid identity. *Sp* and *Sk wtf6* share 82% nucleotide identity, but only 74% amino acid identity. Altogether, the *wtf* loci show much greater sequence divergence than the 99.5% genome average identity between *Sp* and *Sk*. Such rapid evolution is a hallmark of genes involved in genetic conflicts, such as loci involved in causing or suppressing meiotic drive (*Daugherty and Malik, 2012*; *McLaughlin and Malik, 2017*; *Henikoff et al., 2001*).

Intriguingly, the *Sk wtf28* drive gene is also the only one of the six genes we tested that also has a putative alternate start codon in exon two that could be used to make a putative short poison isoform. Additionally, *Hu et al., 2017* also identified two different *wtf* drivers in another *Sp* isolate (CBS5557) and both have a potential alternate start codon in exon 2. Of the 25 *wtf* loci in *Sp*, four (*wtf4*, *wtf13*, *wtf19* and *wtf23*) also appear to be capable of encoding two proteins and we predict these are active drive genes. In contrast, the intact genes we tested that did not confer drive, *Sk wtf2*, *wtf5*, *wtf6*, and *wtf26*, all encode genes similar to the antidote isoform of *Sk wtf4* but appear to lack a shorter poison isoform. Together, our results and those of *Hu et al., 2017*) are consistent with the hypothesis that the ancestral function of the *wtf* family is to confer meiotic drive.

## Discussion

### *Sk wtf4* uses distinct transcripts to encode a meiotic drive system

Our study demonstrates that *Sk wtf4* is a novel, gamete-killing meiotic drive locus. Like the *Spok* family of drivers found in *Podospora*, *wtf4* is an autonomous drive system that confers both the ability to kill gametes that do not inherit the gene and the ability to protect the gametes that do (*Grognet et al., 2014*). We show that *wtf4* achieves these disparate functions by a previously undescribed mechanism in which the gene encodes a poison protein from one transcriptional start site and an antidote protein from an alternative transcriptional start site. We show that the poison protein is trans-acting and has the capacity to destroy all gametes, but that the antidote remains in the gametes that inherit the *wtf4* locus and specifically rescues them from destruction.

The poison-antidote mechanism of *Sk wtf4* is comparable to the bacterial toxin-antitoxin (TA) systems. These systems are found in most prokaryotes and have been extensively studied. TA systems consist of a toxin that will prevent cell growth or viability and an antitoxin that neutralizes the toxin using a wide variety of mechanisms, typically being classified into six different types (*Lee and Lee, 2016*). Interestingly, some toxins are stable, transmembrane proteins that act by disrupting membrane integrity and are counteracted by either an unstable small RNA (*Lee and Lee, 2016*; *Unterholzner et al., 2013*) or a protein that degrades the toxin mRNA (*Wang et al., 2012*). In our poison-antidote meiotic drive system, *Sk wtf4* creates two putative trans-membrane proteins: a trans-acting poison and spore-specific antidote. While the exact mechanism of toxicity of Wtf4[poison] is unknown, we hypothesize that it could be disrupting membrane integrity in a similar manner to the membrane-lytic toxins of some TA systems (*Lee and Lee, 2016*; *Unterholzner et al., 2013*). In contrast, we speculate that Wtf4[antidote] protects the spores that inherited *Sk wtf4* by sequestering the poison for degradation. The spore specificity of Wtf4[antidote] could be due to late translation or a spore retention signal within exon 1, because that is the only region that Wtf4[poison] is lacking. In addition, work by Hu *et al.* suggests that the C-termini of Wtf proteins may be more important for the poison than for the antidote functions, despite both proteins being generated by a single given *wtf* gene sharing a common C-terminus (*Hu et al., 2017*).

Outside of its role in meiotic drive, *wtf4* has no apparent role in promoting fertility (*Figure 2A*, diploid 15). Instead, the gene causes about half of all gametes to be destroyed in heterozygotes. In other words, the wild-type allele of *wtf4 causes* infertility to promote its own fitness. This puts *wtf4* into a state of genetic conflict with the rest of the genome because infertility is clearly bad for fitness of loci unlinked to *wtf4*. Unlinked variants that can suppress drive would be favored by natural selection because they increase fitness (*Crow, 1991*). Novel *wtf4* variants that can evade this suppression to reestablish drive would then be favored. This evolutionary dynamic is analogous to that observed between viruses and host immune systems and is well known to foster a 'molecular arms race' in which both sides must continually innovate (*Daugherty and Malik, 2012*; *McLaughlin and Malik, 2017*). Consistent with the idea that the gene is locked in such an arms race, the DNA sequence divergence between *Sk* and *Sp* at the *wtf4* locus is more than 20-fold higher than the genome-wide average (*Rhind et al., 2011*; *Zanders et al., 2014*).

The evolution of *wtf4* elicits the question of how the gene can rapidly evolve while maintaining specificity between the poison and antidote it encodes. Uncoupling these components leads to sterility, an evolutionary dead end. It is possible that such variants do arise and are quickly purged from populations. We propose, however, that the coding sequence overlap between the poison and antidote could promote specificity between the two components, for example, by the antidote acting as a dominant suppressor of the poison. In this manner, the poison could diverge without losing the self-protection conferred by the antidote. Using a shared sequence to confer specificity between drive components may be a recurring theme amongst gamete-killers. In their analyses of *Podospora Spok* genes, Grognet and colleagues found that they could generate antidote-only alleles and a partial poison-only allele via C- and N-terminal tagging of the proteins, respectively. They were not, however, able to generate separation of function alleles of *Spok1* by making N- or C- terminal deletions (*Grognet et al., 2014*). Future work identifying the molecular mechanisms of both families of drivers will be required to test the idea that the fact that the poison and antidote are encoded by overlapping sequences minimizes the probability of disrupting their specificity.

## Could the expansion and rapid evolution of the *wtf* gene family be a result of meiotic drive?

Combined with our previous work, the varied phenotypes of our *Sp* chromosome 3 introgressions reveal a complex landscape of meiotic drive loci in the *Sk* and *Sp* genomes (*Zanders et al., 2014*). As *Sk wtf4* is a member of the large *wtf* gene family, the most likely candidates underlying these other drive loci are *wtf* genes. For example, there are >20 *wtf* loci on *Sp* (and likely *Sk*) chromosome 3 that could contribute to the complex drive phenotypes we observe (*Figure 1*) (*Bowen et al., 2003*). Consistent with the idea that the *Sk wtf4* is not unique in its ability to drive, we showed that *Sk wtf28* can also cause drive and Hu and colleagues demonstrate that two additional *wtf* genes from another isolate of the *Sp* family also cause meiotic drive (*Hu et al., 2017*).

Although not all *wtfs* are capable of autonomously causing meiotic drive, their rapid evolution is still consistent with their involvement in meiotic drive (*McLaughlin and Malik, 2017*). We propose that different *wtf* genes represent distinct evolutionary stages. The putative ancestral type (*Sk wtf4* and *wtf28*) are still active as meiotic drivers and encode both poison and antidote proteins. The next stratum represents genes (*Sk wtf2, wtf5, wtf6* and *wtf26*) that have lost poison, but not antidote function. As we have shown for the *Sk wtf4^antidote^* allele, such alleles are unlikely to cause meiotic drive as they have lost their poison-coding capacity, but they still have protective function against the ancestral drive allele and thus may have been selectively retained as 'domesticated parasites.' Over time, when the protective function is no longer beneficial and selected for (e.g. if the ancestral drive allele is lost from the population), such antidote genes may also eventually degenerate. Therefore, the final stratum represents putative *wtf* pseudogenes such as *Sk wtf21*, in which both the poison and antidote function have decayed.

There are 25 *wtf* loci in the *Sp* genome and this work combined with that of Hu and colleagues implicates these genes in causing and/or modifying meiotic drive (*Bowen et al., 2003*; *Hu et al., 2017* ). Meiotic drive has therefore played a significant role in the evolution of the *Sp* group of fission yeasts, despite the heavy fitness costs these selfish loci can levy. It is striking to consider how long these selfish genes went undetected in such a simple and intensely studied organism. How many more such parasites are lurking undetected in eukaryotic genomes?

## Materials and methods

### Crosses

For the mapping crosses, fertility and meiotic drive assay, the crosses were carried out similar to the description in *Zanders et al. (2014)*. This required making stable diploids, because many of the strains used are homothallic (h$^{90}$) and their self-mating would generate many non-informative spores. Briefly, we mixed ~200 μL of overnight culture from each haploid parent in a microcentrifuge tube, spun down the cells and plated them on either SPA (1% glucose, 7.3 mM KH$_2$PO$_4$, vitamins, agar) or MEA (3% malt extract, agar) for 12–15 hr at room temperature to allow the cells to mate. We observed no differences in meiotic drive phenotypes for diploids generated on SPA vs. MEA. We generally use SPA, but for some matings have more success isolating stable diploids from MEA. We scraped off the mated cells and spread them on a medium to select heterozygous diploids (generally minimal yeast nitrogen base plates). We grew diploid colonies overnight in 5 mL of rich YEL broth (0.5% yeast extract, 3% glucose, 250 mg/L of adenine, lysine, histidine, leucine, and uracil). We then plated a small amount of the cultures ≤100 μL onto SPA to induce sporulation, and plated a diluted sample onto YEA (same as YEL, but with agar). We screened the colonies that grew on the YEA plate via replica plating to diagnostic media to verify that the culture was comprised of heterozygous diploid cells. If not, the culture was not assayed further. After 3–7 days, we scooped up the mixture of cells, asci, and spores from the SPA plates, treated it with glusulase and ethanol to kill vegetative cells and to release spores from asci, and plated the spores on YEA. We then phenotyped the spore colonies using standard approaches. For some control loci, we could not easily verify heterozygosity in the diploid test described above. For these loci, we verified heterozygosity of the parent diploid in the progeny. If the parent diploid proved not to be heterozygous, the diploid was eliminated. For each cross, we assayed at least two independently created diploids. The number of progeny we scored varied between experiments. To map *Sk wtf4*, we assayed at least 100 viable progeny per cross. To characterize *Sk wtf4*, we assayed at least viable 200 gametes per cross. The one exception was the *Sk wtf4*$^{poison}$/ *Sk wtf4*$^{antidote}$ cross in which we had to assay allele transmission by PCR and sequencing (described below). For that cross, we assayed 50 viable progeny.

### Mapping *Sk wtf4* region

All strains used and their genotypes can be found in *Supplementary file 3*. We deposited sequencing data from all high-throughput sequencing to GenBank accession number PRJNA376152. We chose to map first a drive allele present on *Sk* chromosome 3 via recombination mapping. To eliminate the effects of drivers and gross chromosomal rearrangements from chromosomes 1 and 2, we began the mapping effort using a strain (SZY558) that contains chromosomes 1 and 2 from *Sk*, but in which most of chromosome 3 was derived from *Sp* (the mosaic chromosome illustrated in *Figure 1B* is from SZY558). We could not use a complete *Sp* chromosome 3 because such a strain lacks essential genes due to a translocation between chromosomes 2 and 3 that occurred in the *Sk* lineage (*Zanders et al., 2014*). Sequencing revealed that chromosome 3 in SZY558 was generated by a crossover event between *Sp* and *Sk* chromosomes somewhere between positions 1,804,477 and 1,810,659 on the *Sp* chromosome. The region to the right of this point contains *Sk* alleles and the strain has the *Sk* karyotype (*Figure 1B*).

The generation of SZY558 is described in *Figure 1—figure supplement 1*. We first crossed SZY201 (*Sp*) to SZY208 (*Sk*) to generate SZY239 and SZY247. Although no recombination was expected in this cross because the two parental strains are *rec12Δ*, both SZY239 and SZY247 must contain a recombinant chromosome 2 and/or a recombinant 3 because they inherited non-parental combinations of markers on chromosomes 2 and 3, and the two species karyotypes are incompatible (*Zanders et al., 2014*; *De Veaux et al., 1992*). Such recombinant spores are quite rare (*Zanders et al., 2014*) but were obtained via selection for a nonparental combination of markers on chromosomes 2 and 3 (e.g. His$^+$ hygromycin resistant). Most such selected progeny are chromosome 3 aneuploids, so we then streaked the strains to single colonies to allow them to lose the additional copy of chromosome 3. We crossed SZY239 to SZY247 to generate a strain (SZY382) that contained the recombinant chromosome 3, but also had the *lys1-37* and *his5*$^+$ markers on chromosomes 1 and 2, respectively. The *lys1-37* and *his5*$^+$ markers in this strain were useful for following chromosomes 1 and 2 in a subsequent cross. We transformed SZY382 with a PCR fragment generated with oligos

255 and 256 using plasmid pAG32 as a template to generate a strain (SZY547) with *arg12Δ::hphMX4* (*Goldstein and McCusker, 1999*). The *arg12* locus is on chromosome 3 in *Sk* and in strain SZY547, which has the *Sk* karyotype. We then crossed SZY547 to SZY192 (*Sk*) to generate strain SZY558. The purpose of this cross was to move the recombinant chromosome 3 into a strain background with pure *Sk* chromosomes 1 and 2 (marked with *lys1+* and *his5Δ::natMX4*).

For mapping *Sk wtf4*, we crossed SZY558 to a differentially marked *Sk* strain (SZY210) to generate recombinant haploid progeny (introgression strains) that contained a smaller fraction of chromosome 3 from *Sp* (*Figure 1B*). We used genetic markers (*ura4*, *ade6* and *arg12*) to select only true haploid recombinants for our introgression strains. We mated the introgression strains and *Sk* (SZY196) to generate diploids 1–8 (*Figure 1C*). The mapping scheme was designed such that diploids generated by these matings were homozygous *rec12Δ*, so recombination would infrequently separate the drive allele from the genetic markers used to distinguish the introgression and *Sk* chromosomes (*De Veaux et al., 1992*). We sequenced at least one introgression representing each phenotype we observed amongst these strains and distinguished *Sp* and *Sk* SNPs as in (*Figure 1—source data 1*) (*Zanders et al., 2014*).

SZY565 (the haploid parent that contributed the mosaic chromosome 3 to diploid 1 in *Figure 1C*) is the introgression strain that contains the smallest region of *Sp*-derived DNA, from position 55,555 to 237,572 (*Figure 1—source data 1*). The *Sk* chromosome drove against this introgression in test crosses. We assumed that whatever feature of the *Sp* genome (either the presence of a target of killing or the absence of an antidote to killing) that conferred the sensitive phenotype (i.e susceptibility to being destroyed by the *Sk* driver) must be within that region and, correspondingly, that the *Sk* drive allele must also be within or very near that region. This is because a drive allele that acts to kill gametes that do not inherit it should target the homologous locus or a closely linked site to prevent self-killing. A drive allele that killed gametes that inherit a locus not linked to the drive allele would be an evolutionary dead end because it would kill gametes bearing the drive allele as often as it would kill gametes bearing the competing allele.

To narrow in on the key drive locus, we crossed SZY565 to SZY196 (*Sk*) to get a strain (SZY649) with the same chromosome 3 as SZY565, but with *his5+* rather than *his5Δ::natMX4* on chromosome 2. The *ura4* locus, at position 116,726–115,589 on chromosome 3, is within the *Sp*-derived region. We added an additional marker (*kanMX4*) within the *Sp*-derived region at position 214,491 to generate strain SZY659. To do this, we first generated plasmid pSZB134 which contains ~1 kb of DNA (amplified from *Sp* genomic DNA with oligos 380 and 381) upstream of the target site (214,491) cloned into the BamHI and BglII sites of pFA6a, and ~1 kb of DNA (amplified with oligos 382 and 383) downstream of the target site cloned into the SacI and SpeI sites of the pFA6a (*Wach et al., 1994*). The transformation cassette was released from pSZB134 via NotI digest and used to make SZY659.

We then crossed SZY659 to a differentially marked *Sk* strain (SZY320; *rec12+*) and screened for haploid progeny that had experienced a crossover within the *Sp*-derived region between the *ura4Δ::natMX4* allele from SZY320 and the *kanMX4* allele in SZY659 (*Figure 1D*). We tested nine such haploids by test crossing them to *Sk* (SZY196). Two haploids had an *Sk*-like phenotype in that they showed Mendelian transmission of the *ura4* locus; the other six showed the sensitive phenotype. We genotyped SNPs of the haploids at a few sites within the region to roughly estimate where the recombination event(s) occurred (*Zanders et al., 2014*). Amongst the haploids with the sensitive phenotype, SZY679 and SZY685 have the most *Sk*-derived DNA: they contain *Sp* DNA only between position ~210,000 (between 207,954 and 210,312) to 237,572 (*Figure 1C*, diploid 9, *Figure 1—source data 1*). The two strains with the Mendelian phenotype (SZY684 and SZY686) contain very little *Sp*-derived DNA. The *Sp* DNA begins between positions 210,312 and 214,500 and ends before 215,926. Comparing these two classes suggested that the key drive locus is located between positions ~ 210,000 and 237,572 (but not within the small region surrounding ~214,000). The annotated features of this region include all or part of 10 genes plus one pseudogene in *Sp* (*Figure 1D*).

## Sequencing of the *Sk wtf4* locus

Using oligos MESZ176 and MESZ177, we amplified from *Sk* genomic DNA the region corresponding to the *wtf3+wtf4* locus in *Sp*. The product amplified is at least 1.5 kb smaller than the corresponding product from *Sp*. We then sequenced the PCR product using oligos 557, 560, 565, 566, 567, 568, 569, 570, 595, 597, 598, 599, 601, 602, and 603 and assembled a 2943 bp contig. This sequence has

been deposited to GenBank, accession number KY652738. We did a BLAST search comparing our *Sk* sequence contig to all *Sp* protein sequences and got *Sp wtf13* and *wtf4* as top hits. The *Sk* region contains only one *wtf*-like gene, whereas the *Sp* region has the complete *wtf4* gene and the *wtf3* pseudogene. As the *Sk* gene appears to be orthologous to *Sp wtf4* based on synteny and sequence similarity, we named the gene *Sk wtf4*.

We used the *Sp* PacBio meiotic transcriptome sequences to predict intron-exon boundaries in *Sk wtf4* (*Kuang et al., 2017*). The authors *Kuang et al. (2017)* kindly provided pre-publication access to 'Iso-Seq' consensus isoform sequences. *wtf* genes are not well-represented in the splice isoform summary tables generated for that study due to the very high nucleotide identity between *wtf* paralogs and stringent filtering of multiply-mapping reads. We therefore re-mapped Iso-Seq data to the *Sp* reference genome assembly using GMAP (*Wu et al., 2016*), reporting only alignments with ≥99% identity and covering ≥99% of the length of the isoform sequence, and using the parameter '–suboptimal-score 20' to reduce secondary matches (this parameter choice successfully eliminates cross-mapping between *wtf4* and *wtf13*). We used IGV (*Thorvaldsdottir et al., 2013*) to visualize splice isoforms for each gene. These data reveal a coding sequence that is slightly different from that of the currently annotated *Sp wtf4* gene (http://www.pombase.org/spombe/result/SPCC548.03c). The long form of *Sk wtf4* has six predicted exons and encodes a 337 amino acid protein with 82% amino acid identity to the 366 amino acid protein encoded by *Sp wtf4*. The TMHMM model predicts six transmembrane helices with high probability (>80%) and one with lower probability (<50%) (*Krogh et al., 2001*).

## Generation of *Sk wtf4Δ* mutants

To generate the *Sk wtf4Δ* mutant, we used the CRISPR-Cas9 system after first failing to generate the mutant via the standard homologous recombination approach (*Jacobs et al., 2014*). This system requires the starting strain to be *ura4⁻* and *leu1⁻*. We generated an *Sk* mutant (SZY661) in which *leu1* was replaced with *hphMX4* in strain SZY320. We did this by first cloning a *leu1Δ::hphMX4* cassette (pSZB136). We made this plasmid by first cloning *leu1⁺* (amplified from *Sk* genomic DNA with oligos 413 and 414) into pFA6a cut with NdeI and ClaI and blunted with Klenow fragment of DNA polymerase I. This new vector was then cut with ClaI and NdeI (within *leu1*) and blunted with Klenow: the *hphMX4* cassette liberated from pAG32 with PvuII and ClaI was ligated into the gap (*Goldstein and McCusker, 1999*). Oligos 413 and 414 were used again to amplify the *leu1Δ::hphMX4* cassette for transformation.

To generate plasmid pSZB184, which encodes a guide RNA targeting the *Sk wtf4* region, we annealed oligos 577 and 578 and cloned them into the CspCI site of pMZ283 (*Jacobs et al., 2014*) We used overlap-PCR to generate a repair cassette containing ~1 kb of homology upstream and downstream of the *Sk wtf4* region flanking the *kanMX4* cassette from pFA6a (*Wach et al., 1994*). We stitched together the products of PCRs generated with oligos 571 and 572, 575 and 576, and 573 and 574 to make the repair cassette. We then transformed strain SZY661 with pMZ222, pSZB184, and the repair cassette. We screened through Ura⁺ Leu⁺ transformants containing both plasmids for *wtf4* deletions via PCR and sequencing. We found that strain SZY862 contained a deletion of *wtf4*, but unexpectedly was not resistant to G418. Sequencing of the region revealed a truncation of the *kanMX4* gene. SZY863 contains the same deletion as SZY862, but is Ura⁺ due to retention of the *ura4⁺* cassette from pSZB184 at an unknown location closely linked to the endogenous *ura4* locus, although the strain retains the *ura4Δ::natMX4* allele at the endogenous *ura4* locus.

## Generation of the *ade6*-targeted constructs

We first generated pSZB188, a plasmid containing the *kanMX4* selectable marker and a mutant *ade6* allele that has 5', central, and 3' deletions. This vector can be cut with KpnI within the mutant *ade6* gene and then integrated into *ade6⁺* to generate Ade⁻ G418-resistant transformants. Other genes can be added to the vector to introduce them into the genome at the *ade6* locus. To construct pSZB188, we first made a mutant *ade6* cassette via overlap PCR stitching a PCR product made from oligos 588 and 589 to one made from oligos 591 and 590. This *ade6* cassette was then digested with BamHI and XhoI and cloned into the BamHI and SalI sites of pFA6a (*Wach et al., 1994*). We cloned the *Sk wtf4* region into pSZB188 by first amplifying the region with oligos 619 and 620. We digested the PCR product with SacI and cloned it into the SacI site of pSZB188 to generate

pSZB189. We introduced KpnI-digested pSZB189 into yeast and selected transformants on YEA with G418 plates. We picked red colonies, as proper integrants should harbor a mutant *ade6* allele flanking the sides of the plasmid sequence. The duplicated *ade6* gene makes the locus unstable and Ade$^+$ revertants that have 'popped out' all plasmid-derived sequences are readily obtained.

The *Sk wtf4-GFP* allele was made using overlap PCR. We amplified the promoter region from *Sk* genomic DNA using oligos 633 and 604 and the open reading frame sequence using oligos 605 and 606. We used pKT127 as a template to amplify yEGFP using oligos 607 and 634 (*Sheff and Thorn, 2004*). We then stitched the three PCR products together using overlap PCR. We cut the resulting cassette with SacI and cloned it into the SacI site of pSZB188 to generate pSZB204. This construct was integrated at *ade6* as described above.

For the *Sk wtf4$^{antidote}$* allele, using overlap PCR, we stitched together two PCR products generated with oligo pairs 735 and 686, and 620 and 736, both using pSZB189 as a template. We cloned the stitched PCR product into the SacI site of pSZB188 to generate pSZB246. We then cut pSZB246 and introduced it into yeast as described above.

We generated the *Sk wtf4$^{poison}$* allele using overlap PCR. *Sk wtf4* has two in-frame start codons in the annotated exon 1. Mutating the first start codon had no phenotype (data not shown), so we mutated both. To mutate the first start codon, we used overlap PCR to stitch together two PCR products made by oligo pairs 701 and 686 and 620 and 702; both reactions used pSZB189 as a template. The stitched PCR product was cloned into the SacI site of pSZB188 to generate pSZB244. We used pSZB244 as a template to mutate the second start codon via overlap PCR. We stitched together PCR fragments generated by oligo pairs 620 and 739 and 686 and 740 and cloned that product into the SacI site of pSZB188 to generate pSZB258. We cut pSZB258 and introduced it into yeast as described above.

We cloned the *Sk mCherry$^{antidote}$-wtf4* allele using overlap PCR. First, we purchased from IDT (Coralville, IA) a synthetic double-stranded DNA gene block including the *Sk wtf4* promotor, the mCherry coding sequence (*Hailey et al., 2002*), five glycine codons, and the first part of *Sk wtf4* exon 1. We amplified that fragment with oligos 620 and 604 and stitched that PCR product to another that contained the rest of the *Sk wtf4* gene amplified with oligos 605 and 687 from plasmid pSZB189. We then cloned that product into the SacI site of pSZB188 to generate pSZB248, which we cut and introduced into yeast as described above.

For the *Sk wtf4$^{poison}$-GFP* allele, we amplified the 5' end of the gene with oligos 620 and 739 using plasmid pSZB244 as a template. We amplified the 3' end of the gene with oligos 740 and 634 using pSZB203 as a template. We then used overlap PCR to stitch those PCR fragments together and cloned the product into the SacI site of pSZB188 to generate pSZB257, which we cut and introduced it into yeast as described above.

We used the same strategy to integrate other *Sk wtf* genes into *Sp*. We used *Sk* genomic DNA as a template to amplify *wtf21+wtf26* with oligos 643 and 644, *wtf2* with oligos 647 and 648, *wtf5* with oligos 649 and 650, and *wtf6+wtf28* with oligos 651 and 652. We cut each cassette with SacI and cloned them into the SacI site of pSZB188 to generate: pSZB209 (*wtf21+wtf26*), pSZB212 (*wtf2*), pSZB217 (*wtf5*), and pSZB215 (*wtf6+wtf28*). We subcloned *Sk wtf6* and *Sk wtf28* from pSZB215 by first amplifying the individual genes using oligo pairs 732 and 652, and 651 and 733, respectively. We then cloned the genes into the SacI site of pSZB188 to generate pSZB252 (*wtf6*) and pSZB254 (*wtf28*). All sequences of these genes have been deposited in GenBank, accession numbers KY652739-KY652742. These constructs were all integrated at *ade6* as described above.

## Assaying allele transmission in *wtf4$^{poison}$/ wtf4$^{antidote}$* diploid

Because the alleles *wtf4$^{poison}$* (SZY1051) and *wtf4$^{antidote}$* (SZY1110) are marked with the same drug marker, to score transmission of alleles for this cross, we used sequencing. We generated diploids and spores as described above. We then plated the spores on YEA, picked the colonies to a YEA master plate and replicated to score control markers. We also prepared lysates for PCR from the master plate by scraping cells off the master plate into 20 μl of 20 mM NaOH. We boiled the cells for 5 min, froze them in liquid nitrogen, boiled again for 10 min, and spun the debris down. Using the supernatant lysate, we amplified the *wtf4* region using oligos A01112 and 678. We then sequenced the exon 1 region using oligo 861. We analyzed the exon 1 region for the start codon mutations mentioned above (*Figure 4A*). If the exon 1 mutations were present, we quantified this as a poison allele; if not present, as an antidote.

## Introducing *ade6+* at *his5*

To avoid *ade6-* mutant auto-fluorescence in cytology, we introduced *ade6+* at the *his5* locus. We amplified a region upstream of *his5* to generate piece A using oligo pair 795 and 796. We amplified a region downstream of *his5* to generate piece C using oligo pair 797 and 798, and to amplify *ade6+* we generated piece B using oligo pair 799 and 800. We stitched together pieces A, B and C using oligo pair 795 and 798 and introduced the product into yeast.

## Cytology

For the fertility assay, we added 5–10 µl of propidium iodide (PI, 1 mg/ml) to 50 µl of $H_2O$, and scraped the yeast from the SPA plate into the PI mix. We then incubated the yeast plus PI mixture at room temperature for 20 min. We took the images on a Zeiss (Germany) Observer.Z1 wide-field microscope with a 40x (1.2 NA) water-immersion objective and collected the emission onto a Hamamatsu ORCA Flash 4.0 using µManager software. We acquired the PI images with BP 530–585 nm excitation and LP 615 emission, using an FT 600 dichroic filter.

For all other fluorescence microscopy, we imaged on a LSM-700 AxioObserver microscope (Zeiss), with a 40x C-Apochromat water-immersion objective (NA 1.2), with 488 and 555 nm excitation. We collected GFP fluorescence through a 490–55 nm bandpass filter and mCherry fluorescence through a 615-nm longpass filter. The continuously variable secondary dichroic filter was positioned at 578 nm. We also imaged using a LSM-780 (Zeiss) microscope, with a 40x C-Apochromat water-immersion objective and 100x alpha Plan-Apochromat oil-immersion objective (NA 1.2 and 1.46, respectively), in photon-counting channel mode with 488 and 561 nm excitation. We collected GFP fluorescence through a 481–552 bandpass filter and mCherry through a 572 longpass filter. For all images acquired on the LSM-780 (Zeiss) microscope, using the same objectives as described above, we also imaged in photon-counting lambda mode, with 488 and 561 nm excitation. We collected fluorescence emission over the entire visible range. After acquisition, we linear unmixed the images using an in-house custom written plugin for ImageJ (https://imagej.nih.gov/ij/). Unmixing was achieved using spectra obtained from control cells. We did unmixing to verify that there was no auto-fluorescence in the cells (*Figure 5—figure supplement 1*) we scored. We did not score auto-fluorescent cells. Brightness and contrast is not the same for all images. We assayed at least 35 asci (but usually >100) for each genotype represented in *Figures 3* and *5*.

## Acknowledgements

We are grateful to T Levin and J Zeitlinger for comments on the manuscript, M Miller for assistance with the figures, Li-Lin Du for sharing results prior to publication, and three thoughtful reviewers (including Nick Rhind and Michael Lichten) for guiding improved data presentation. Original data underlying this manuscript can be accessed from the Stowers Original Data Repository at http://www.stowers.org/research/publications/libpb-1132. This work was supported by NIH R01 GM031693 and R35 GM118120 (GRS); NIH R01 GM74108, The Mathers Foundation, and HHMI (HSM); The Stowers Institute for Medical Research (SLJ and SEZ); and NIH K99/R00 GM114436 (SEZ).

## Additional information

### Competing interests

NLN, MABN, HSM, SEZ: Inventor on Patent application of based on this work. Patent application serial 62/491,107. The other authors declare that no competing interests exist.

### Funding

| Funder | Grant reference number | Author |
| --- | --- | --- |
| National Institutes of Health | R35 GM118120 | Gerald R Smith |
| National Institutes of Health | R01 GM031693 | Gerald R Smith |
| Stowers Institute for Medical Research | | Sue L Jaspersen Sarah E Zanders |

| G Harold and Leila Y. Mathers Foundation | | Harmit S Malik |
| Howard Hughes Medical Institute | | Harmit S Malik |
| National Institutes of Health | R01 GM74108 | Harmit S Malik |
| National Institutes of Health | K99/R00 GM114436 | Sarah E Zanders |

The funders had no role in study design, data collection and interpretation, or the decision to submit the work for publication.

## Author contributions

NLN, MABN, MTE, Data curation, Formal analysis, Validation, Investigation, Visualization, Writing—review and editing; JMY, Formal analysis, Investigation, Writing—review and editing; JJL, Formal analysis, Investigation, Methodology, Writing—review and editing; JSY, Data curation, Formal analysis, Investigation, Writing—review and editing; GRS, Conceptualization, Formal analysis, Supervision, Investigation, Methodology, Project administration, Writing—review and editing; SLJ, Supervision, Project administration, Writing—review and editing; HSM, Conceptualization, Supervision, Funding acquisition, Project administration, Writing—review and editing; SEZ, Conceptualization, Data curation, Formal analysis, Supervision, Funding acquisition, Validation, Investigation, Visualization, Methodology, Writing—original draft, Project administration, Writing—review and editing

## Author ORCIDs

Janet M Young, http://orcid.org/0000-0001-8220-8427
Harmit S Malik, http://orcid.org/0000-0001-6005-0016
Sarah E Zanders, http://orcid.org/0000-0003-1867-986X

## Additional files

### Supplementary files

• Supplementary file 1. Raw data of allele transmission from *Figures 2–5*. Each horizontal entry represents the relevant genotype and allele transmission of the indicated diploids. The first column represents the diploid number, which matches the numbers shown in *Figures 2–5*. In columns 2–5, the two haploid parent strains (SZY#s) are indicated as are the alleles contributed by those parent strains at the experimental locus monitored for drive. Alleles derived from *Sp* are green whereas those from *Sk* are purple. For diploids 11–15, transmission of the *wtf4* locus was followed using alleles of *ura4*, which is fortuitously closely linked to *wtf4*. Columns 9 and 10 indicate which phenotypes were followed at the drive loci and the number of progeny that displayed each phenotype. Some progeny inherited both alleles at a given drive locus and when the markers were codominant we could detect those disomes. The number of those disomes, which are likely heterozygous diploids or aneuploids, are shown in column 11 and their overall frequency is shown in column 12. If we did not have codominant markers, columns 11 and 12 are filled with —. Column 13 is a measure of meiotic drive. It shows the fraction of the non-disomic progeny that inherited allele 1 (column 3). Column 14 shows the total number of progeny assayed for each diploid and column 15 is the p value calculated from a G-test comparing the allele transmission of allele 1 to a control. Diploid 13 served as the control for diploids 11, 12, 14 and 15. Diploid 16 served as the control for the rest of the diploids. Columns 6–8 are internal controls for each diploid. These controls represent an additional heterozygous locus unlinked to the meiotic drive locus that should undergo Mendelian allele transmission. The *lys* locus is *lys4*, the *ade* locus is *ade6*, and the *ura* locus is *ura4*. The final column indicates the number of independently generated diploids that were tested for each genotype.

• Supplementary file 2. Raw data for PI-staining phenotypes from *Figures 2–5*. Each horizontal entry represents the relevant genotype and allele transmission of the indicated diploids. The first column represents the diploid number, which matches the numbers shown in *Figures 2–5*. In columns 2–5, the two haploid parent strains (SZY#s) are indicated as are the alleles contributed by those parent strains at the experimental locus monitored for drive. Alleles derived from *Sp* are green whereas those from *Sk* are purple. Columns 6 and 7 indicate the number of spores that stained with PI (dead

spores) and those that did not (likely living spores) and column 8 shows the percentage of spores that did not stain with PI. Column 9 shows the number of stained asci that were scored for each diploid type. Spores not contained within 4-spore asci were not scored. Column 10 shows the p value from a G test comparing the number of stained and unstained spores for each diploid to a control diploid. Diploid 13 served as a control for diploids 11, 12, 14 and 15. Diploid 16 served as a control for all other diploids. The number of independently generated diploid strains that were tested is indicated in the last column.

• Supplementary file 3. Yeast strains.

• Supplementary file 4. Plasmids.

• Supplementary file 5. Oligos.

## Major datasets

The following dataset was generated:

| Author(s) | Year | Dataset title | Dataset URL | Database, license, and accessibility information |
|---|---|---|---|---|
| Nuckolls NL, Bravo Núñez MA, Eickbush MT, Young JM, Lange JJ, Yu JS, Smith GR, Jaspersen SL, Malik HS, Zanders SE | 2017 | Fission yeast meiotic drive recombination mapping | https://www.ncbi.nlm.nih.gov/bioproject/PRJNA376152/ | Publicly available at the NCBI BioProject database (accession no: PRJNA376152) |

The following previously published dataset was used:

| Author(s) | Year | Dataset title | Dataset URL | Database, license, and accessibility information |
|---|---|---|---|---|
| Kuang Z, Boeke JD, Canzar S | 2016 | The dynamic landscape of fission yeast meiosis alternative-splice isoforms | https://www.ncbi.nlm.nih.gov/geo/query/acc.cgi?acc=GSE79802 | Publicly available at the NCBI Gene Expression Omnibus (accession no: GSE79802) |

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
