## [Decision Letter]

Thank you for submitting your article “*wtf* genes are prolific dual poison-antidote meiotic drivers" for consideration by *eLife*. Your article has been favorably evaluated by Diethard Tautz (Senior Editor) and three reviewers, one of whom is a member of our Board of Reviewing Editors. The following individuals involved in review of your submission have agreed to reveal their identity: Nick Rhind (Reviewer #2) and Michael Lichten (Reviewer #3).

The reviewers have discussed the reviews with one another and the Reviewing Editor has drafted this decision to help you prepare a revised submission.

Summary:

This article reports analysis of a gene family that shows meiotic drive behavior. It's an extension of a previous paper by the senior author, who demonstrated the existence of meiotic drive elements in *S. pombe/S. kambucha* hybrids, but did not characterize these elements in detail. The current paper (and an accompanying paper from Hu et al.) identifies at least a subset of these drive elements as belonging to the *wtf* gene family, a group of repeated genes that tend to be found near transposable elements or their LTR remnants. This paper focuses on one of these genes, *wtf4*, and shows that the *S. kambucha* allele encodes a protein with two isoforms, a short form that encodes a spore-killing toxin that is expressed in meiotic cells before spore formation, and a long form that encodes an antidote that is expressed in a spore-specific manner. Further analysis identifies a second *wtf* locus in *S. kambucha, wtf28*, with similar gene structure and meiotic drive behavior. This, in combination with an accompanying paper (Hu et al.), provide a convincing case for the existence of multiple single-locus, toxin/antidote-type meiotic drive elements in the *Schizosaccharomyces* clade. Because of the novelty of this system, and because this work provides the first mechanistic insight into how a single-locus drive element can work, the paper is of considerable interest.

Essential revisions:

While the science in this paper is really exciting, the writing and presentation are sloppy and obtuse-it often feels as if the authors are having a conversation with themselves, and are focused on presentation rather than communication (these are *not* the same thing). This makes it a frustrating paper to read at more when one has an in-depth interest. More seriously, underlying experimental data are often needlessly absent or obscured.

1) Figure 1. One really has to hunt to figure out what the introgression diploids really are or how they were made. Since this is an online journal, why not illustrate their construction, and the chr 3 configuration within each diploid? This figure, like many others, uses terminology that is distant from the actual data. For example, "% aneuploid or diploid" "gametes" (Figure 1) is not what was assayed – which was the fraction of viable spores that contain both parental chr 3 markers (but what were those markers?) and is thus a subset of possible disomic/diploid progeny (think uniparental disomy)?

In addition, part of the reason for the high level of chr 3 "disomy" among viable spores is because these are *rec12-/rec12*- diploids, which normally have a high degree of chr3 "disomy". This should be clearly communicated, and an Sk/Sk control (with suitably marked chrs 3) included to show the base level of chr 3 "disomy/diploidy" in *rec12*- mutants in the absence of any drive.

Another example – "% transmission Sk" – actually the fraction viable spores that only display one parental set of chr 3 markers, that display the "Sk" markers – but what were those markers? It's almost impossible to figure this out. Finally, data underlying Figure 1 are not presented at all, and need to be included in [Supplementary-material SD4-data] (that is, once [Supplementary-material SD4-data] is modified to make it clear and accessible, see below).

2) Figure 2, [Supplementary-material SD4-data]. The term "% fertility" is a misnomer. What is assayed is spores that fail to exclude PI; it's not even spore viability, since there are many ways a spore could not give rise to viable progeny yet still exclude PI. If authors can show that the vast majority of spores that exclude π give rise to colonies, then "fertility" is OK, but otherwise an operative term (i.e. "PI excluding") should be used.

3) [Supplementary-material SD4-data]. This table should include data from Figure 1 (see above). It absolutely requires a legend that explains what each column means (for example, N means 2 different things) and how the numbers were determined.

4) Figure 5. RNAseq data underlying the identification of two *wtf4* transcripts should be presented.

5) Subsection “Expansion and rapid evolution of the *wtf* family is consistent with a role in meiotic drive”. Data and methods underlying the conclusions about the other *wtfs* that were tested should be presented, as these are important in vivo tests of the hypothesis. If diploids similar to diploid 9 in Figure 2 were made for the other *wtfs*, that data would be sufficient. It would also be nice (although not essential) if the authors could make a supplementary file that aligns all of the *wtfs* examined, to show where the two start sites are or are not. Since the RNAseq data for all the Sp *wtfs* are available, it would be interesting to know if the "tamed" *wtfs* only make the longer antidote mRNA, or if they still express both mRNAs. It would also be useful to ask which of the *wtfs* identified by Hu et al. (i.e. *cws* in CBS5557) are extant toxin-antidote systems using the dual start-site criterion, and which, if any, have degenerated to antidote-only; the same request will be made of Hu et al., so perhaps the two sets of authors could coordinate on this. It would also be useful to mention that Hu et al. have identified the C-termini of two *wtf/cws* loci in CBS5557as being important for toxin but not for antidote function.

6) There is a long history of toxin/antidote system in bacteria, which has been extensively reviewed. Adding a brief discussion of this to the Introduction or Discussion would add perspective to the article, and might offer some insight into possible *wtf* mechanisms.

7) Given that what the authors describe are gamete / spore killers, wouldn't it be more accurate to refer to the process as "drive" instead of as "meiotic drive" (following the recommendation of Burt & Trivers from their Genes in Conflict book – see page 16)? The phenomenon described has nothing to do with meiosis.

8) The authors need to be more sophisticated in their use of the concept of species. *S. kambucha* was so named on the basis of hybrid sterility <https://www.ncbi.nlm.nih.gov/pubmed/12399374> but is actually itself a mosaic of two strains of *S. pombe* (represented by 972 and NCYC132) which show greater sequence divergence from each other than either do from *S. kambucha* <https://www.ncbi.nlm.nih.gov/pubmed/21511999>. Thus, none of the three populations are genetically isolated on an evolutionary timescale. Therefore, be any modern definition of the term, *S. pombe* 972, *S. pombe* NCYC132 and *S. kambucha* are not separate species.

9) It would be of great interest to readers to have some speculation of how a putative trans-membrane protein could function in a toxin-antidote system and how the toxin comes to be distributed differently than the antidote.

---

## [Author Response]

*Essential revisions:*

*While the science in this paper is really exciting, the writing and presentation are sloppy and obtuse-it often feels as if the authors are having a conversation with themselves, and are focused on presentation rather than communication (these are not the same thing). This makes it a frustrating paper to read at more when one has an in-depth interest. More seriously, underlying experimental data are often needlessly absent or obscured.*

We have noted the reviewer concerns about writing and presentation, and have endeavored to correct these deficiencies through extensive rewriting and redrafting of figures in our revision.

*1) Figure 1. One really has to hunt to figure out what the introgression diploids really are or how they were made. Since this is an online journal, why not illustrate their construction, and the chr 3 configuration within each diploid?*

We agree with this suggestion. We have rewritten the section of the paper describing the introgression diploids to describe them in more detail. We also added Figure 1—figure supplement 1 to describe the generation of the strain we used to generate the introgressions. We also expanded Figure 1 to illustrate how the introgression strains were made (Figure 1). These changes should make clear what the introgression diploids are and how they were made.

This figure, like many others, uses terminology that is distant from the actual data. For example, "% aneuploid or diploid" "gametes" (Figure 1) is not what was assayed – which was the fraction of viable spores that contain both parental chr 3 markers (but what were those markers?) and is thus a subset of possible disomic/diploid progeny (think uniparental disomy)?

*In addition, part of the reason for the high level of chr 3 "disomy" among viable spores is because these are rec12-/rec12- diploids, which normally have a high degree of chr3 "disomy". This should be clearly communicated, and an Sk/Sk control (with suitably marked chrs 3) included to show the base level of chr 3 "disomy/diploidy" in rec12- mutants in the absence of any drive.*

*Another example – "% transmission Sk" – actually the fraction viable spores that only display one parental set of chr 3 markers, that display the "Sk" markers – but what were those markers? It's almost impossible to figure this out.*

We agree with these points. Indeed, there is a higher degree of ch3 disomy in *rec12-/rec12-* diploids; we now display this *rec12-/rec12-* control data in the table (diploid 10) to point this out explicitly, rather than just using it for statistical comparison. We also agree with the reviewers’ complaints about the terminology – we have changed the labels we used on the tables in Figure 1 and elsewhere in the paper where similar data are displayed. We also changed the table in Figure 1 to make it more clear which markers we assayed.

*Finally, data underlying Figure 1 are not presented at all, and need to be included in [Supplementary-material SD4-data] (that is, once [Supplementary-material SD4-data] is modified to make it clear and accessible, see below).*

We agree, and omission of these data was an oversight. These data are found in the new [Supplementary-material SD2-data], which includes a detailed figure legend.

*2) Figure 2, [Supplementary-material SD4-data]. The term "% fertility" is a misnomer. What is assayed is spores that fail to exclude PI; it's not even spore viability, since there are many ways a spore could not give rise to viable progeny yet still exclude PI. If authors can show that the vast majority of spores that exclude π give rise to colonies, then "fertility" is OK, but otherwise an operative term (i.e. "PI excluding") should be used.*

We agree with this point as well. “% fertility” was an imprecise way to describe our PI assay and we have changed that to “% PI-excluding” in the text and figures. The PI assay works well for spores killed by meiotic drive (because they essentially pop), but we did not intend to present it as a general fertility assay. We have also added a new table ([Supplementary-material SD3-data]) that compares% PI-excluding cells to a more standard measure of fertility in fission yeast, the viable spore yield assay. We note that the two measures are largely congruent so this assay is a good surrogate for overall spore viability in our crosses.

*3) [Supplementary-material SD4-data]. This table should include data from Figure 1 (see above). It absolutely requires a legend that explains what each column means (for example, N means 2 different things) and how the numbers were determined.*

We agree. We have removed the old version of [Supplementary-material SD4-data] because it was unclear and ambiguous. The raw data underlying Figure 1 are now found in [Supplementary-material SD2-data], which has a detailed legend. We have kept this raw data separate from the rest of the raw genetics data (found in new [Supplementary-material SD4-data] and [Supplementary-material SD5-data]) to help make clear that these strains are *rec12-*, while the rest of the data are from *rec12+* strains.

*4) Figure 5. RNAseq data underlying the identification of two wtf4 transcripts should be presented.*

We agree. A summary of these data is now presented as Figure 3—figure supplement 1. We also expanded our description in the Methods of how these data were analyzed.

5) Subsection “Expansion and rapid evolution of the wtf family is consistent with a role in meiotic drive”. Data and methods underlying the conclusions about the other wtfs that were tested should be presented, as these are important in vivo tests of the hypothesis. If diploids similar to diploid 9 in Figure 2 were made for the other wtfs, that data would be sufficient

We now present these methods and data in our revision.

*It would also be nice (although not essential) if the authors could make a supplementary file that aligns all of the wtfs examined, to show where the two start sites are or are not. Since the RNAseq data for all the Sp wtfs are available, it would be interesting to know if the "tamed" wtfs only make the longer antidote mRNA, or if they still express both mRNAs.*

We agree that these are interesting questions, but we have not added this information as we believe it is beyond the scope of this paper. It is not a straightforward comparison to make as the *wtf* gene family is very rapidly evolving. We are currently undertaking detailed bioinformatic analyses of the *wtf* gene family and its remarkable evolution, with the intention of presenting this analysis in the near future.

*It would also be useful to ask which of the wtfs identified by Hu et al. (i.e. cws in CBS5557) are extant toxin-antidote systems using the dual start-site criterion, and which, if any, have degenerated to antidote-only; the same request will be made of Hu et al., so perhaps the two sets of authors could coordinate on this. It would also be useful to mention that Hu et al. have identified the C-termini of two wtf/cws loci in CBS5557as being important for toxin but not for antidote function.*

Consistent with our model, both of the driving *wtf*s identified by Hu et al. appear to have alternate start sites within exon 2, similar to *Sk wtf28*. We now mention this as additional support for our model in the paper. We also mention Hu et al.’s result regarding the function of the C-terminus.

*6) There is a long history of toxin/antidote system in bacteria, which has been extensively reviewed. Adding a brief discussion of this to the Introduction or Discussion would add perspective to the article, and might offer some insight into possible wtf mechanisms.*

We have added a section on these bacterial systems to the Discussion. We note that only the general evolutionary principles of toxin-antidote systems are conserved and relevant to the *wtf* genes. Even among bacteria, they are myriad means to accomplish inheritance via the toxin/antidote mechanism. We therefore point to recent reviews where this topic and diversity has been extensively discussed.

*7) Given that what the authors describe are gamete / spore killers, wouldn't it be more accurate to refer to the process as "drive" instead of as "meiotic drive" (following the recommendation of Burt & Trivers from their Genes in Conflict book – see page 16)? The phenomenon described has nothing to do with meiosis.*

We disagree with this point but acknowledge lack of a scientific consensus regarding what to call ‘meiotic drive’. Although *wtf* genes do not participate in meiotic drive as originally envisioned to occur in asymmetric (female) meiosis (Sandler and Novitski, 1957), this term has, however, long been accepted to describe genetic elements that bias allele transmission by acting after the meiotic divisions by some of the original pioneers in this field (Zimmering et al., 1970): *“*Meiotic drive has been defined by Sandler & Novitski as any alteration of the normal process of meiosis with the consequence that a heterozygote for two genetic alternatives produces an effective gametic pool with an excess of one type; such a pattern of behavior will drastically alter the frequency of alleles in a population in such a way that a driven allele may increase in frequency in spite of deleterious physiological effects. This general concept has, however, been taken to include transmissional anomalies that are not strictly meiotic, but with similar populational consequences [see, for example, Lewontin]; this extended meaning seems justified and thus the more general definition will be used in this review.”The *wtf* genes have as much to do with ‘meiotic drive’ as other mechanisms of post-meiotic dysfunction such as the SD-Responder system in *Drosophila* and the t-locus in mice, which have been classically referred to as meiotic drive systems in the literature. We, therefore, favor the term ‘meiotic drive’ for gamete-killers in general and *wtf*s specifically.

We also disagree with the comment that the phenomenon has nothing to do with meiosis. Indeed, one could imagine that toxin-antidote systems could be manifest in mitosis in eukaryotes just as they are in prokaryotes. However, *wtf* systems specifically exploit the kinetics of alternate, dual expression in meiosis to manifest their action. Indeed, *wtf* genes are highly transcribed in meiosis and we know that the *wtf* poison is present in cells prior to the completion of the meiotic divisions. Given that we do not fully understand the molecular mechanisms of these genes and their evolutionary implications, it seems premature to say they have no effect on meiosis.

*8) The authors need to be more sophisticated in their use of the concept of species. S. kambucha was so named on the basis of hybrid sterility <https://www.ncbi.nlm.nih.gov/pubmed/12399374> but is actually itself a mosaic of two strains of S. pombe (represented by 972 and NCYC132) which show greater sequence divergence from each other than either do from S. kambucha <https://www.ncbi.nlm.nih.gov/pubmed/21511999>. Thus, none of the three populations are genetically isolated on an evolutionary timescale. Therefore, be any modern definition of the term, S. pombe 972, S. pombe NCYC132 and S. kambucha are not separate species.*

Here again, we disagree with the reviewers. *S. kambucha* and *S. pombe* are reproductively isolated and fit the Ernst Mayr definition of ‘biological species’. The fact that they are a mosaic of two strains of *S. pombe* is interesting but inconsequential to the fact that they are reproductively isolated. We also point to famous examples of ring species (e.g. butterflies), many of which are themselves a result of hybridization, in which reproductive isolation is not a fully commutative phenotype.

Although not within the purview of our paper, we note that many other isolates grouped under the umbrella label “*S. pombe*” have clear reproductive barriers. There are many routes to generating reproductive barriers and thus distinct biological species and there is no set timescale during which these changes occur. What is remarkable about the *S. pombe* isolates is the fast rate at which reproductive boundaries have arisen.

*9) It would be of great interest to readers to have some speculation of how a putative trans-membrane protein could function in a toxin-antidote system and how the toxin comes to be distributed differently than the antidote.*

We agree, and we have now added our hypothesis in the Discussion section.